



# Connecting flow-topography interactions, vorticity balance, baroclinic instability and transport in the Southern Ocean: the case of an idealized storm track

5 Julien Jouanno[1], Xavier Capet[2]

[1]LEGOS, Université de Toulouse, IRD, CNRS, CNES, UPS, Toulouse, France
[2]CNRS-IRD-Sorbonne Universités, UPMC, MNHN, LOCEAN Laboratory, Paris, France

Corresponding author: julien.jouanno@ird.fr

**Abstract.** The dynamical balance of the Antarctic circumpolar current and their implications on the functioning of the world ocean are not fully understood and poorly represented in global circulation models. In this study, the sensitivities of an idealized Southern Ocean (SO) storm track are explored with a set of eddy-rich numerical simulations. The classical partition between barotropic and baroclinic modes is sensitive to current-topography interactions in the mesoscale range 10-100 km, as 15 comparisons between simulations with rough or smooth bathymetry reveal. Configurations with a rough bottom have weak barotropic motions, no wind-driven gyre in the lee of topographic ridges, less efficient baroclinic turbulence, and thus larger circumpolar transport rates. The difference in circumpolar transport depends on the strength with which (external) thermohaline forcings by the rest of the world ocean constrain the stratification at the northern edge of the SO. The study highlights the need for a comprehensive treatment of the Antarctic Circumpolar Current (ACC) interactions with the ocean floor. It also sheds some 20 light on the behavior of idealized storm tracks recently modelled: i) the saturation mechanism, whereby the circumpolar transport does not depend on wind intensity, is a robust and generic attribute of ACC-like circumpolar flows ii) the adjustment toward saturation can take place over widely different time scales (from months to years) depending on the possibility (or not) for barotropic Rossby waves to propagate signals of wind change and accelerate/decelerate SO wind-driven gyres. The real SO with a typical ACC saturation time scale of 2-3 years seems to lie in the "rough bottom/no wind-driven gyre" regime.

## 1. Introduction

The strength of the Antarctic Circumpolar Current (ACC) is controlled at first order by a balance between the eastward momentum imparted by the persistent Southern Ocean (SO) winds and the topographic form stress at the ocean bottom (Munk and Palmén 1951, Hugues and de Cuevas 2001). The bulk of the bottom pressure gradients is thought to be provided by the major submarine 30 ridge (Kerguelen Plateau, Macquarie Ridge, Scotian Arc, and East Pacific Rise) and the South America continent (Munk and Palmén 1951, Gille 1997, Masich et al. 2015).

Along with their decelerating action on the mean flow, the major ridges result in strong inhomogeneity of the SO dynamics. Indeed, they act to concentrate and energize the eddy activity downstream of the topography, in regions often referred as "storm tracks". The underlying process is a local intensification of the baroclinicity and baroclinic instability of the flow



(Bischoff and Thompson 2014, Abernathey and Cessi 2014, Chapman et al. 2015). Localized baroclinic instability goes in hand with a suppression of eddy growth away from the ridge (Abernathey and Cessi 2014). Overall, ridges profoundly shape the SO dynamics, stratification (Abernathey and Cessi 2014, Thompson and Naveira Garabato 2014) as well as subduction hot spots (Sallée et al. 2010).

Another potentially important aspect of the dynamics through which ridges affect the SO circulation was recently uncovered by Nadeau and Ferrari (2015). In the presence of topographic ridges, their idealized numerical simulations of the ACC reveal the co-existence of the classical circumpolar flow and closed recirculating gyres driven by Sverdrup like dynamics. Further, increasing wind intensity leads to increasing gyre circulation without modification of the circumpolar transport, suggesting that the saturation of the circumpolar transport with increasing winds may be connected with gyre dynamics.

Apart from the major ridges (Figure 1), the sea floor is shaped by topographic features with horizontal scales from hundreds of meters to tens of kilometers (mainly abyssal hills) which are though to dissipate most of the large scale wind power input in the SO through the generation of internal lee waves (Nikurashin and Ferrari 2011) and to provide high abyssal mixing (Nikurashin and Ferrari, 2010). But a substantial fraction of the bottom topography variance is also contained at scales in between the major ridges (100 km and larger) and the typical width scale of the abyssal hills (O 0.1-10 km; see Goff and Jordan 1988, Nikurashin and Ferrari 2010b). This range of topographic scale between 10 and ~100km will be referred to as "mesoscale" and is
in part associated with the abyssal hills (e.g. see Fig. 1 from Goff and Arbic 2010)[1].

Naveira Garabato et al. (2013) showed that the effectiveness of topography in arresting the time mean ocean circulation decreases as the horizontal dimension of the topographic features decreases. Thus, small scale topography is not expected to exert a first order control on the SO dynamics, as is the case for the major ridges. Tréguier and McWilliams (1990) analyzed ACC channel dynamics with random topography with different wavenumber spectra. Their results point to the role of large scale
topography in providing the overwhelming fraction of the ACC form stress while smaller scale topography (wavelength < 500 km in their case) has little effect on the transport.

Using periodic channel simulations with a meridional ridge, this study investigates the sensitivity of an idealized SO storm track to the presence or absence of mesoscale topographic irregularities, a case that has not been investigated in Tréguier and McWilliams (1990). Our results show that the form drag exerted by the mesoscale topography has an indirect but significant
influence on the ACC transport. Disentangling the cause of this indirect influence sheds some light on the key processes that set the dynamical balance of the SO. The numerical experiments are presented in Section 2. Section 3 presents the dynamical balance of the SO in a channel simulation with a ridge and investigates the consequences of adding rough topography. Processes at play are discussed in Section 4 and concluding remarks are given in Section 5.

**2. Model**

**2.1 The numerical set-up**

---

[1] Abyssal hills frequently have a length-width aspect ratio of 5 or more [e.g. Goff and Arbic 2010).



The numerical set up consists of a periodic channel configuration of 4000 km long (Lx, zonal direction) and 2000 km wide (Ly, meridional direction), with walls at the northern and southern boundaries. It is inspired from the simulations described in Abernathey et al. (2011) and Abernathey and Cessi (2014), and aims to represent a zonal portion of the SO (Figure 2).

The numerical code is the oceanic component of the Nucleus for European Modeling of the Ocean program (NEMO, Madec 2014). It solves the three dimensional primitive equations discretized on a C-grid and fixed vertical levels (z-coordinate). Horizontal resolution is 5-km. There are 50 levels in the vertical, with 10 levels in the upper 100 meters and cell thickness reaching 175 m near the bottom. Precisely, the thickness of the bottom cells is adjusted to improve the representation of the bottom topography, with a partial step thickness set larger than 10% of the standard thickness of the grid cell. The model is run on a β-plane with $f_0=1.10^{-4}$ s$^{-1}$ at the center of the domain and $\beta= 1.10^{-11}$ m$^{-1}$ s$^{-1}$. A 3rd order upstream biased scheme (UP3) is used for

both tracer and momentum advection, with no explicit horizontal diffusion. The vertical diffusion coefficients are given by a Generic Length Scale (GLS) scheme with a k-ε turbulent closure (Reffray et al. 2015). Bottom friction is linear with a bottom drag coefficient of 4 10$^{-4}$ m s$^{-1}$ and is computed based on an explicit formulation. The free surface formulation is linear and uses a filtered free surface scheme (Roullet and Madec, 2000). We use a linear equation of state with temperature as the only state variable and a thermal expansion coefficient $\alpha=2.10^{-4}$ K$^{-1}$. The temporal integration involves a modified Leap Frog Asselin Filter, with a

coefficient of 0.1 and a time step of 400 seconds.

The forcing consists in an eastward wind defined as:

$$u_{10} = U_0 \sin\left(\frac{\pi y}{L_y}\right)$$

$u_{10} = U_0 \sin\left(\frac{\pi y}{L_y}\right)$, with $u_0$=10 m s$^{-1}$. The wind stress is calculated using the formulation from Large and Yeager (2009) formulation. This leads to a maximum wind stress of 0.14 N m$^{-2}$ at Ly/2 and zero wind stress curl at the northern and southern

walls.

At the northern boundary, the model can be restored toward an exponential temperature as motivated by observations (Karsten and Mashall 2002) and following the formulation proposed in Abernathey et al. (2011):

$$T_{north}(z) = \Delta T * \left( e^{z/h} - e^{H/h}\right)\left(1 - e^{z/h}\right)$$

with $\Delta T$=8°C, H=4000 m the depth of the domain, and h=1000m. The relaxation coefficient varies linearly from 0 at y=1900 km

to 7 day$^{-1}$ at Ly.

The surface heat flux $Q_{air-sea}$ is built using a relaxation method toward a prescribed sea surface temperature (SST) climatology. It depends on a sensitivity term $\gamma$ set to 30 W m$^{-2}$ K$^{-1}$ (Barnier et al. 1995) and on the difference between $T_{model}$ and a predefined climatological SST field $T_{clim}$:

$Q_{air-sea} = \gamma (T_{clim} - T_{model})$, with $T_{clim}(y) = \Delta T * y/L_y$ .

Simulations have been performed with two types of topography. All include a Gaussian shaped ridge centered in middle of the domain (x=2000 km). The height of the meridional ridge is given by $h_0 e^{-x^2/\sigma^2} - H$, with $h_0$=2000 m the maximum height of the ridge, σ=75 km and H=4000m the maximum depth of the domain.

**2.2 Sensitivity to bottom roughness and northern restoring**





We performed two sets of simulations that differ by their bathymetry outside the ridge (Figure 2b,c). In R+F (for ridge & flat)

simulations, the bottom floor outside the ridge is flat and located at 3500 m depth (hrms=0m). In R+R (for ridge & rough) simulations, the bottom depth outside the ridge varies between 3000 and 4000 m depth with random fluctuations of horizontal wavelength between 10 and 100 km (h$_{rms}$=250m) and the constraint that the averaged depth remains 3500m as in R+F. The bottom roughness, defined as the variance of the bottom height H, is 6.2 10$^4$ m$^2$. This choice of roughness and horizontal scales is consistent with the characteristics of some SO topography but not for all the sectors (Figure 1; see also Wu et al. 2011 or Fig. 10 in Goff

2010). The impact of spatially varying roughness is not addressed in this study and would deserve dedicated sensitivity experiments to connect finely with the dynamics of the real Southern Ocean.

Restoring temperature toward a prescribed stratification profile at the northern boundary exerts a strong constraint on the model solution and can partially account for the influence of low latitude and northern hemisphere ocean sectors. Channel configurations with limited meridional extent have alternatively been using strategies with or without northern restoring (e.g.

Abernathey et al. 2011, Abernathey and Cessi 2014). In order to test the sensitivity of our results to this constraint, we performed additional experiments without restoring, which are referred as R+Fnr and R+Rnr.

The four simulations (R+F, R+R, R+Fnr and R+Rnr) are initialized with the same initial conditions consisting of an ocean at rest and a stratification given by the stratification prescribed at the northern boundary. They are integrated over 150 years. Unless otherwise stated, monthly instantaneous fields from the last 10 years are used for diagnostics. In addition, simulations with

increased wind forcing (maximum wind stress of 0.28 N m$^{-2}$ at Y=Ly/2) have been integrated for 30 years, starting from the equilibrium state of the reference set of simulations (forced with maximum wind stress of 0.14 N m$^{-2}$ as mentioned above).

**2.3 Vorticity balance**

The sensitivity of "Sverdrup"-like dynamics in our set of simulations is investigated through the analysis of the different terms of the model barotropic vorticity (BV) equation (e.g., Jackson et al. 2006). The BV equation reads as follows:

$\beta V = J(pb, H) + \boldsymbol{k}.\nabla \times (\boldsymbol{\tau_w}) - \boldsymbol{k}.\nabla \times (\boldsymbol{\tau_b}) + \boldsymbol{k}.\nabla \times (\boldsymbol{A}),$

with β the derivative of the planetary vorticity, V the vertically integrated meridional velocity, $J(p_b,H)$ the bottom pressure torque, with $p_b$ the pressure at the sea floor and H the ocean depth, and $\boldsymbol{k}.\nabla \times (\boldsymbol{A})$ the non-linear advection term. The different terms were evaluated by taking the curl of the vertically integrated momentum balance terms computed on-line. The contributions of the lateral and temporal diffusion are very weak and are not shown.

**3. Results**

**3.1 Overall characteristics of the simulations**

Independently of bottom roughness or northern restoring, the topographic ridge forces a large-scale standing meander in its lee (Figure 3a), as found in previous studies (Abernathey and Cessi 2014, Nadeau and Ferrari 2015, Chapman et al. 2015). The baroclinic instability of the meander, as revealed by the distribution of vertical eddy buoyancy flux (Figure 3c; $< \overline{w'b'} >$ with w'

and b' the vertical velocity and buoyancy anomalies with respect to time averaged values, $<\cdot>$ the 0-500 m vertical averaging, and $^{-}$ the 10 years temporal averaging), energizes the eddy field in the 500 km downstream of the ridge (Figure 3b). Further downstream, the energy of the eddies drops off (Figure 3b). The resulting EKE distribution is typical of SO storm tracks as described for example in Chapman et al. (2015). The kinetic energy of the mean flow (MKE, Figure 4a), the eddy kinetic energy





(EKE, Figure 4b), and the marked isopycnal slope in the upper 1000 meters (Figure 5c), all illustrate the strong baroclinic character

of the dynamics.

The equilibrium state of the different simulations is almost achieved after 100 years as indicated by the stabilization of the zonal transport and total kinetic energy (Figure 5a,b)[2]. The topographic drag of the ridge constrains the zonal flow to transports between 25 and 75 Sv, values which are relatively weak compared to the barotropic transport obtained for similar channel simulations with flat bottom (~800 Sv in Abernathey and Cessi 2014) or rough topography only (~300 Sv in Jouanno et al. 2016).

**3.2 Sensitivity to bottom roughness**

We now focus on the influence of bottom roughness on the channel dynamics. We do this by comparing simulations R+F and R+R. We start with the simulations including northern restoring because their sensitivity to topography is easier to understand, the northern restoring acting on the density structure so the mean state does not depart too much between the two simulations. The key result is that bottom roughness leads to a ~60% increase of the zonal transport (Figure 5b). These changes of zonal transport are

associated with profound modifications of the overall dynamics.

**a)  Vertical structure of the flow**

The vertical structure of both steady and turbulent flow components is very sensitive to bottom roughness. In the latitude range where the ACC is located the flow is impacted (above 700 m) as revealed by values of MKE and EKE that are larger in R+F compared to R+R (Figures 4a,b). Most importantly, finite values of MKE and EKE persist below 1000 meters in R+F (Figures

4a,b), indicative of a significant contribution of the barotropic mode, while MKE and EKE are vanishingly small in the deep layers of R+R. This agrees with the modal decomposition carried out at y=1000 km (Figure 6). The barotropic mode contains most of the energy in R+F, while the energy in R+R is almost evenly distributed between the barotropic and first baroclinic mode.

The spectral analysis in Figure 7 highlights the profound differences between the two solutions. Near the ocean floor up to ~1500 m depth, bottom roughness energizes the flow at scales finer than $\sim 1/(2.10^{-4}\ \mathrm{m}^{-1})$. On the other hand, it is responsible

for a marked reduction of deep ocean KE at scales larger than $\sim 1/(2.10^{-4}\ \mathrm{m}^{-1})$, *i.e.*, both at large scale and mesoscale. This directly affects the flow up to 1500 meters, i.e., at depth well above the bottom floor (Figure 7b).

The changes in the vertical structure of the flow can be interpreted as follows: bottom roughness forces zero flow at the bottom and thus weaken the barotropisation process for both the large and mesoscale dynamics. This echoes recent findings by LaCasce (2017) that bathymetric slopes promote surface intensified modes.

**b)  Storm track intensity**

More locally, dynamics in the lee of the ridge is largely affected by bottom roughness. The comparison between R+F and R+R shows that bottom roughness reduces the zonal extent of the standing meander (Figure 8a) and the EKE levels in the lee of the

---

[2] The total KE and the transport takes more time to equilibrate in R+R compare to R+F (Figure 5a), with a large KE increase in the first 10 years of spin-up and a KE decrease in the following ~90 years. This slow decrease of the domain averaged kinetic energy in R+R between years 10 and 100 is related to a slow destratification of the southernmost part of the domain (the weak stratification of the final state can be seen in Figure 5c). As indicated in Section 2, the simulation is initialized using a stratified density profile. During the first years of the simulation there is enough background stratification to sustain the existence of baroclinic eddies over the entire domain. The subsequent uplifting of the isopycnal and associated destratification in the south progressively prevents the existence of baroclinic eddies, while barotropic eddies cannot develop due to the strong constrain exerted by the bottom form drag. At equilibrium, the region located between Y=0 and ~500km is devoid of eddies. The time taken for this sequence to unfold may explain the slower transport equilibration in this simulation.





ridge (Figure 8b). This is associated with a weakening of the local baroclinic instability conversion, as revealed by weaker vertical eddy buoyancy flux in R+R in the vicinity of the ridge and meander (Figure 8c). The response is distinct in the rest of the domain

where bottom roughness leaves EKE levels approximately unchanged (Fig. 8b) and even slightly increases EKE (Figure 8a). Note that this enhancement of EKE arises despite weaker baroclinic instability conversion in the upper 500 m (Figure 8c). Thus, rough topography limits the eddy energy at the location of the stationary meanders but also favors the persistence of the eddy energy far from the ridge.

### c) Strength of the gyre mode

Wind-driven gyres in the lee of tall topographic ridges are a potential attribute of the SO circulation that has received recent attention (Nadeau and Ferrari 2015) but remains poorly known, including observationally. Essential to make progress is to understand the ACC response to wind stress curl input and balance of vorticity. In R+F the barotropic streamfunction (Figure 9a) reveals the presence of closed recirculating gyres in the lee of the meridional ridge, consistent with the double gyre circulation found by Nadeau and Ferrari (2015) in the presence of a tall ridge. When rough topography is added, the southern (resp. northern)

gyre completely (resp. nearly) disappears (Figure 9c and also 9e where the meridional structure of the zonal flow in the lee of the ridge is represented).

To help interpret this result, the barotropic vorticity balance in R+R and R+F averaged between y=400 and 600 km is shown in Figure 10 for two portions of the zonal domain: an area under direct influence of the ridge (between x=1500 and 2500km) and an area including the rest of the zonal domain. In the range of latitude considered, the western boundary current forming at the

ridge location is northward and well defined in R+F (Figure 9). This is reflected in the BV balance by the negative and large values of the term -βV (Figure 10a). At first order, this northward flow is balanced by the bottom pressure torque. In the rest of the domain (Figure 10b), pressure torque is no more effective (because the bottom is flat) and the wind stress curl is balanced by a southward barotropic flow. This is the classical wind-driven gyre balance (Munk 1950; Hugues 2005, Nadeau and Ferrari 2015) whose relevance to the real SO remains uncertain as mentioned above.

The vorticity balance is fundamentally different when rough topography is included. First, the northward barotropic flow at the ridge location present in R+F is absent (Figure 10c). The bottom pressure torque there mainly acts to balance the local wind stress curl. In the rest of the domain, the vorticity balance is similar to that occurring at the ridge: a large fraction of the wind stress curl is balanced by bottom pressure torque, limiting both the southward transport and the influence of the bottom friction.

### 3.3 Sensitivity to northern restoring

In R+F and R+R, the joint action of the air-sea heat fluxes and eddy buoyancy fluxes set the interior stratification and large scale dynamical equilibrium of the ACC. The restoring of the density field toward a specified profile at the northern boundary can be seen as an additional thermohaline constraint that prevents an equilibration of the two solutions in widely different states. We now compare this set of simulations with restoring at the northern boundary (simulations R+F and R+R) to a similar one without the restoring (simulations R+Fnr and R+Rnr). Simulations without restoring may be thought of as idealized representations of an ocean

where the SO dynamics dictates hydrographic conditions north of the ACC path to the rest of the world ocean (though with the remaining constraint that the residual overturning circulation be zero). Conversely, simulations with restoring would represent conditions in which the rest of the world ocean imposes a fixed stratification at the northern edge of the SO. Each is a limit case distinct from the real ocean where significant water mass transformation occurs in the SO with large rates of water volume import/export by the meridional overturning cells.





200       Most of our previous results are not qualitatively dependent on the choice of restoring the northern stratification. Specifically, adding rough bathymetry without northern restoring still: increases the ACC transport (Figures 5b,e); decreases deep MKE and EKE (Figure 4a,b); weakens the vertical buoyancy flux in the lee of the ridge (Figure8f) although only slightly with no restoring; and strongly affects the BV balance in such a way that wind-driven gyres are present (resp. absent) in smooth (resp. rough) bottom conditions (Figure 9). Two important distinctions are noteworthy. First, the ACC transport sensitivity is far greater

without northern restoring (~170% increase from 23Sv in R+Fnr to 62Sv in R+Rnr). Second, bottom roughness strongly decreases total KE when restoring is applied while total KE is very weakly affected when no restoring is applied. We attribute this to the fact that the more efficient release of available potential energy in the absence of rough bathymetry (Figure 4c) can significantly modify the ACC thermohaline structure in the simulations without restoring whereas it cannot when tightly constrained by the restoring (compare the departures between isotherms in Figs.5c and f). Further elaboration is provided in Section 4.

**3.4 Sensibility to wind stress increase**

Sensitivity to wind intensity is explored by doubling the wind stress forcing for all simulations previously used. In agreement with the dominant theory (e.g. Meredith and Hogg 2006, Morrison and Hogg 2012), all the configurations respond with an increase of the total kinetic energy (Figure 11a) but exhibit a saturation of the zonal transport (Figure 11b). In R+F and R+Fnr, the saturation is accompanied by a strengthening of the recirculating gyre (Figure 9e,f), as observed in Nadeau and Ferrari (2015). In presence

of rough topography, the weak gyre circulation previously found in the northern part of the domain intensifies slightly. In the south, close examination of Figures 9e,f reveals that the barotropic streamfunction develops weak maxima near y = 500 km for doubled wind intensity. The tendency to form wind-driven gyres is minor though and occurs while the nature of the BV balance remains unchanged (not shown).  Most of the additional wind stress curl is balanced by bottom pressure torque in and out of the ridge area, as opposed to meridional Sverdrup transport. This result questions the recent interpretation of the transport saturation mechanism

placing emphasis on the coexistence of a gyre mode together with the circumpolar flow (Nadeau and Ferrari 2015; see Section 4).

       On the other hand, the transient response to wind increase in the presence and absence of bottom roughness are distinct in important ways. In Figure 12 we present the time series of circumpolar transport and EKE for R+F and R+R. Insets provide enhanced details for the period where the solutions adjust to the sudden wind intensity doubling at t=150 years. Adjustments were monitored with outputs at monthly frequency which limits our ability to determine short time scales precisely. More importantly,

a difficulty arises from the fact that the temporal changes following the wind increase combine a deterministic response and stochastic variability. Large ensemble of simulations would be needed to disentangle the two components and we limit ourselves to a qualitative description of the main differences between R+F and R+R. The EKE adjustment in R+R occurs over a time period of ~4 years  and roughly conforms to the descriptions made in Meredith and Hogg (2006) and the general expectations drawn from eddy saturation theories. In R+F, the EKE adjustment is comparatively much faster. It is nearly completed after 6 months, except

for a small downward trend during 10-20 years that follows a slight initial overshoot.

No transport adjustment is discernible in R+F. Conversely, R+R undergoes a noticeable transport adjustment. An initial transport increase of about 8 Sv occurs over the first few months. The subsequent time period of about 15-20 years exhibits a trend toward smaller transports. Toward year 165 the circumpolar transport has finally returned to the steady state with values a few Sverdrups below those prior to the wind increase. Note that the initial spin-up of R+R also includes a secondary adjustment period between

years 60 and 100 (Fig. 5) which is absent in R+F and R+Rnr.





The reasons underlying the adjustment differences between R+R and R+F are examined in the context of the saturation theory in Section 4.

## 4. Discussion

### 4.1 Dynamical interpretation of the bottom roughness effect

This part of the discussion is an attempt to hold and connect together (in words) the issue of flow-topography interactions (1) and their consequences (in cascade order) on the barotropic component of the flow (2), the BV balance (3), baroclinic instability and the storm track dynamics in the vicinity of the ridge (4), and finally the ACC transport (5)[3]. Starting from (1) we remind that the impact of bottom topography on the general circulation and how it responds to atmospheric forcings has been studied for a long time (e.g., Munk and Palmén 1951, Tréguier and McWilliams 1990, Hughes and De Cuevas 2001, Ward and Hogg 2011). In our

simulations $h_{rms}$ is large enough for the f/h potential vorticity field to be dominated by numerous closed isolines. In this situation, the barotropic component of the flow is strongly affected (Tréguier and McWilliams 1990, Hugues and Killworth 1997, LaCasce 2010). Specifically, barotropic Rossby waves are no longer permitted (Anderson and Killworth 1979, LaCasce 2017). In the context of closed basins, wind-driven gyres and a Sverdrup balance are nonetheless being established in the upper ocean by the baroclinic Rossby waves (Anderson and Killworth 1979). However, this is not possible in the context of the ACC where baroclinic Rossby

wave propagation is too slow compared to advection by the mean flow. As a consequence, only in the flat bottom configuration can the Sverdrup balance emerge.

Beside the effect on Rossby wave modes, the barotropic circulation is greatly diminished in the presence of rough bathymetry (Figure 7), and so is the strength of the deep circulation (Figure 9). Our interpretation is that the presence of the topography inhibits or counteracts (Trossman et al. 2017) the barotropization process generally associated with turbulent

geophysical flows. Horizontally integrated energy budgets carried out for different depth layers of fluid provide support to this interpretation[4]. In R+F, pressure work is a term of dominant importance in the flow energetics. It transfers KE vertically from the upper ocean (0-1500 m depth) into the deep ocean (3000 - 4000 m depth). The magnitude of the transient KE transfer into the deep layer (computed with buoyancy, pressure and velocity anomalies respective to zonally averaged values) is reduced by a factor over 3.5 in the presence of rough bottom, i.e., the barotropization mechanism is greatly hampered. In turn, the slowdown of the deep

---

[3] In search for an alternative and possibly simpler interpretation one reviewer suggested that the the transport sensitivities revealed by this study may be the consequence of vertical stratification differences between our simulations (in our primitive equation framework the stratification cannot be held fixed unless artificial restoring is employed). Everything else being unchanged the ACC transport tends to increase with stratification (e.g., in the quasi-geostrophic simulations of Nadeau and Ferrari 2015). In contrast, we find that stratification is generally stronger in R+F (resp. R+Fnr) than in R+R (resp. R+Rnr). For instance, the stratification averaged over the subdomain 500 km < y < 1500 km (the central part of the domain where the zonal flow is intensified) and -3000 m < z < 0 (the part of the water column above the topographic hills) is ~ 15% stronger in R+F than in R+R. Thus, the stratification differences cannot be invoked to explain that larger transport values found in R+R than in R+F.

[4] A different interpretation may be proposed in the context of surface modes decomposition (LaCasce, 2017). Surface mode decomposition explicitly accounts for the presence of variable bathymetry in the vertical mode decomposition which suppresses the barotropic mode.



circulation has important consequences on the flow-ridge interaction whose ability to produce topographic form stress is severely reduced (compare on-ridge magnitude of the pressure torque for R+F and R+R in Fig. 11).

Overall, the differences in flow-topography interactions and their consequences on the barotropic circulation (turbulent flow and linear Rossby wave mode) yield fundamentally different bottom form stress and BV balances. The distribution of bottom form stress is relatively uniform zonally in solutions with rough bottom. Conversely, large bottom form stresses are confined east

in the lee of ridges in solutions with smooth bottom, in conjunction with the presence of intensified boundary currents. The BV balance and boundary currents then resemble those typical of wind-driven gyres (Nadeau and Ferrari 2015; Figure 10).

The circulation pattern resulting from the interaction between an ACC-like flow and a ridge (the so-called "standing wave response" in Abernathey and Cessi 2014) is responsible for intense frontogenesis, Available Potential Energy (APE) release, and eddy heat fluxes in the lee of the ridge. In the same sector, simulations with smooth bottom produce boundary currents which

combine to the standing wave response, and further enhance the frontogenetic tendency and the overall ability of the storm track to release APE, thereby acting to flatten the isopycnals and limit the ACC transport.

The reduced baroclinicity and zonal transport in R+F and R+Fnr can thus be seen as the manifestation of the boundary current effect on local baroclinic instability in the lee of the ridge. In the simulations without restoring this manifestation on baroclinic instability is less evident because the mean thermohaline structure of the ACC has significantly more freedom to adjust

in response to the strength of baroclinic instability processes. In turn, this response of the mean state lead to a negative feedback by modulating the intensity of baroclinic processes which ends up being quite similar with and without rough bathymetry in the absence of northern restoring (compare EKE and APE release rate for R+Fnr and R+Rnr in Figures 4c and 8f).

Overall, the surprising transport sensitivity that motivated this study reveals important upscaling effects resulting from mesoscale flow-topography interactions. They corroborate the finding of Nadeau et al. (2013) in a quasi-geostrophic framework

that the ACC transport increases when the realism of flow-topography interactions is improved. Our work contributes to its interpretation and strives to unravel the underlying causal chain of processes.

The beginning of this research developed with the hypothesis that R+F and R+R differed by the characteristics of their dominant mode of baroclinic instability and a stronger (resp. weaker) local instability mode in R+F (resp. R+R). Here, local instability mode refers to the definition proposed by Pierrehumbert (1984). The concept of local instability mode is used by

Abernathey and Cessi (2014) to rationalize the behavior of a simulation resembling R+F. The onset of gyres and associated boundary currents when the ocean floor is smooth certainly makes local baroclinic instability modes growing in the vicinity of the ridge stronger. Given the specifics of local instability developments we might thus expect to see a lesser tendency for flow perturbations in R+R to remain quasi-stationary in the vicinity of the ridge (Pierrehumbert 1984; Abernathey & Cessi 2014). Hovmoeller diagrams for surface temperature perturbations in R+R and R+F show no particular evidence of this (Figure 13). Also

note that R+R has lower baroclinic conversions rates than R+F not just about the ridge but also far outside its range of influence. A simple and general dynamical explanation for the baroclinic instability sensitivity to bottom roughness revealed in this study would be that rough topography upsets the subtle coupling between fluid layers required for baroclinic instability perturbations to grow by constraining the mean and time-variable flow.

### 4.2 Implications for the eddy saturation process

Baroclinic instability, which is the main source of energy for the mesoscale eddy field in the SO consumes the APE imparted by wind-driven upwelling. It occurs in such a way that additional energy input by the wind enhances EKE but leaves APE and ACC



transport nearly unchanged. This is the so-called eddy saturation effect which limits the sensitivity of the circumpolar transport to changes in the wind forcing magnitude (Morrison and Hogg 2012, Munday et al. 2013, Marshall et al, 2017). Overall, our findings confirm the robustness of the saturation process with respect to major changes in model configuration, which translate into varied

baroclinic instability regimes/efficiency (as previously noted in Nadeau et al. 2013) and a wide range of ACC transports. In particular, the saturation process is more generic than the study by Nadeau and Ferrari (2015) suggests. The work of Nadeau and Ferrari (2015) highlights the role of the gyre mode and Sverdrup balance in the saturation mechanism. To the contrary, in our study the effectiveness of the saturation process (e.g., measured as the long-term relative change in ACC transport when doubling the wind intensity) is insensitive to the presence or absence of a wind-driven gyre component in the SO.

305         In Nadeau and Ferrari (2015), increasing the bottom drag coefficient reduces the intensity of the gyre circulation and also impedes the ACC transport saturation. Bottom roughness and bottom drag are sometimes thought to be interchangeable ways to boost the topographic control over oceanic flows (Arbic and Flierl 2004, LaCasce 2017). As anticipated by Nadeau and Ferrari (2015), this is not the case with respect to the saturation process whose efficiency is not affected by bottom roughness whereas increased bottom drag reduces the intensity of the gyre circulation and also impedes the ACC transport saturation in Nadeau and

Ferrari (2016). We attribute this to the fact that large bottom drag produces a non-physical damping of the turbulent flow and changes the nature of the momentum and vorticity balances (we recall that bottom form stress is not a drag force - Tréguier and McWilliams (1990) - and, in particular, provides no sink in the energy budget).

        Recently, Sinha and Abernathey (2016) have offered important insight into the transient behavior of an ACC system subjected to wind changes. Following wind intensification, saturation is the final outcome of a process involving two stages: a

rapid build up of APE (and ACC transport increase) followed by a slower buildup of EKE which feeds back onto baroclinic instability efficiency and allows APE (and ACC transport) to return back to (or near) their initial levels. Time scales needed for saturation to act on R+F and R+R turn out to be markedly different. Most interestingly, R+F has an almost immediate equilibration of EKE levels to wind changes and no transient effect on ACC transport can be noticed at the monthly temporal resolution we used to track simulation spin-ups. The response time of R+R is of the order of a few years, in line with typical values reported by

previous studies. Following up on the dynamical discussion in Section 4a we interpret the rapid adjustment in R+F described in Section 3 as follows: barotropic Rossby waves with phase speeds of a few m/s adjust the interior Sverdrup transport to new wind conditions in about 10 days (i.e. the time scale to travel across the entire domain); adjustment of the compensating boundary transport on the eastern side of the ridge follows a somewhat slower but comparable pace (Anderson and Gill 1975); density advection by the boundary current locally modifies frontogenetic conditions on time scales of weeks (advection is slower than barotropic Rossby wave propagation but meridional distances to be covered by advection are smaller than the zonal scale of the

system); EKE responds on time scales ~ weeks typical of baroclinic instability growth (Tulloch et al. 2011) and *locally* provides the additional APE release and lateral heat fluxes necessary to prevent APE and circumpolar transport to increase. In R+R barotropic Rossby waves are not permitted and a much slower baroclinic adjustment process of diffusive nature unfolds as described in Sinha and Abernathey (2016). The response of EKE in R+F is faster than typically estimated in many observational

(Meredith and Hogg 2006, Morrow et al. 2010) or realistic modelling studies (Meredith and Hogg 2006; Langlais et al. 2015) but this remains a subject of debate (Wilson et al. 2014). Recent numerical experiments (Patara et al. 2016) indicate that the correlation between wind and EKE underlying the eddy saturation mechanism are sensitive to the regional level of bottom roughness. In this context, we hypothesize that the main topographic obstacles in the SO delimit a small number of sectors whose dynamics includes





a degree of gyre circulation that depends on the small/meso-scale bathymetry. Realistic SO simulations that differ in their bottom
roughness would be instructive to examine this hypothesis.

**5. Conclusions**

The comparison between different numerical simulations for a reentrant zonal jet revealed that the baroclinicity of the flow is sensitive to current-topography interactions in the mesoscale range 10-100 km, with large consequences on the zonal and gyre transport.

340         Using semi-realistic simulations of the SO, this study investigates the influence of bottom roughness on the dynamics of an idealized ACC type flow. While relying on a limited number of simulations our analyses offer important insight into the sensitivities of ACC model representations. A key ingredient impacting the ACC dynamics is the presence of tall obstacles that provide support for form drag and bottom pressure torque. The main sensitivity explored herein concerns more complex flow-topography interactions and more specifically the role of "random" rough bathymetry combined to a tall ridge. Bottom roughness

(with $h_{rms}$ of 250m, typical of abyssal hills) is found to have profound consequences on the ACC equilibration. Specifically, it damps the barotropic mode which has major implications on the momentum and barotropic vorticity balances. In turn, this affects the efficiency of baroclinic instability processes at releasing APE and limit the circumpolar transport.

        Overall, our study points to the importance and sensitivity of current-topography interactions in the mesoscale range (10-100 km) for the dynamics of the ACC. The question of whether the real ocean is in a regime that is more aptly described by our

rough or smooth simulation remains to be elucidated. From a modeling perspective, the bottom roughness considered in this study enters in a scale range of bottom topography which is unequally resolved by climate or global circulation models at resolution between ¼° and 1°. Recent efforts have been dedicated to parameterizing energy dissipation and mixing caused by the abyssal hills (Nikurashin et al. 2010b, De Lavergne et al. 2016). To our knowledge the impact of subgrid-scale topographic drag has, on the other hand, been forsaken in ocean modelling. Our results advocate for a systematic and scale-dependent exploration of flow-

topography interactions so that the transfer of momentum due to bottom form stress are realistically represented irrespective of the unresolved bottom roughness. A starting point is available in atmospheric sciences where approaches have been developed to parameterize sub-grid scale orographic drag (e.g. Lott and Miller 1997).

**Acknowledgements**

This study has been supported by IRD and CNRS and has been founded by the French ANR project SMOC. Supercomputing
facilities were provided by GENCI projects GEN7298 and GEN1140. A special thanks to J. Chanut for discussion and his assistance with the model set-up.

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





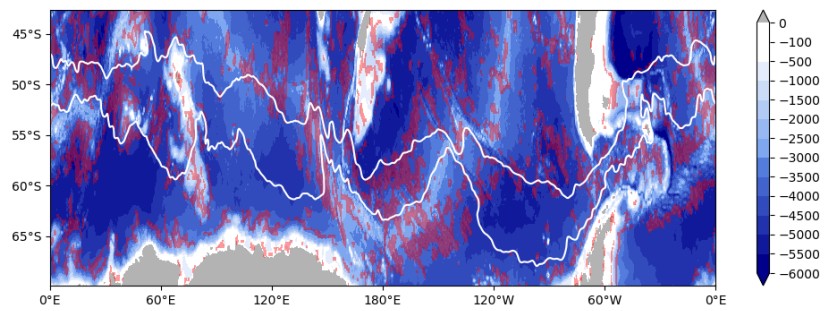


**Figure 1.** Topography [m] of the Southern Ocean from ETOPO2 (National Geophysical Data Center, NOAA). The red dashed areas indicate areas with a topographic roughness (computed as the variance of the topography over an area of 100x100km as in Wu et al. 2011) between $3.10^4$ and $10^5$ $m^2$. The main pathway of the ACC is identified by two isocontours [0.4 and 1 m] of mean

dynamical topography from AVISO.


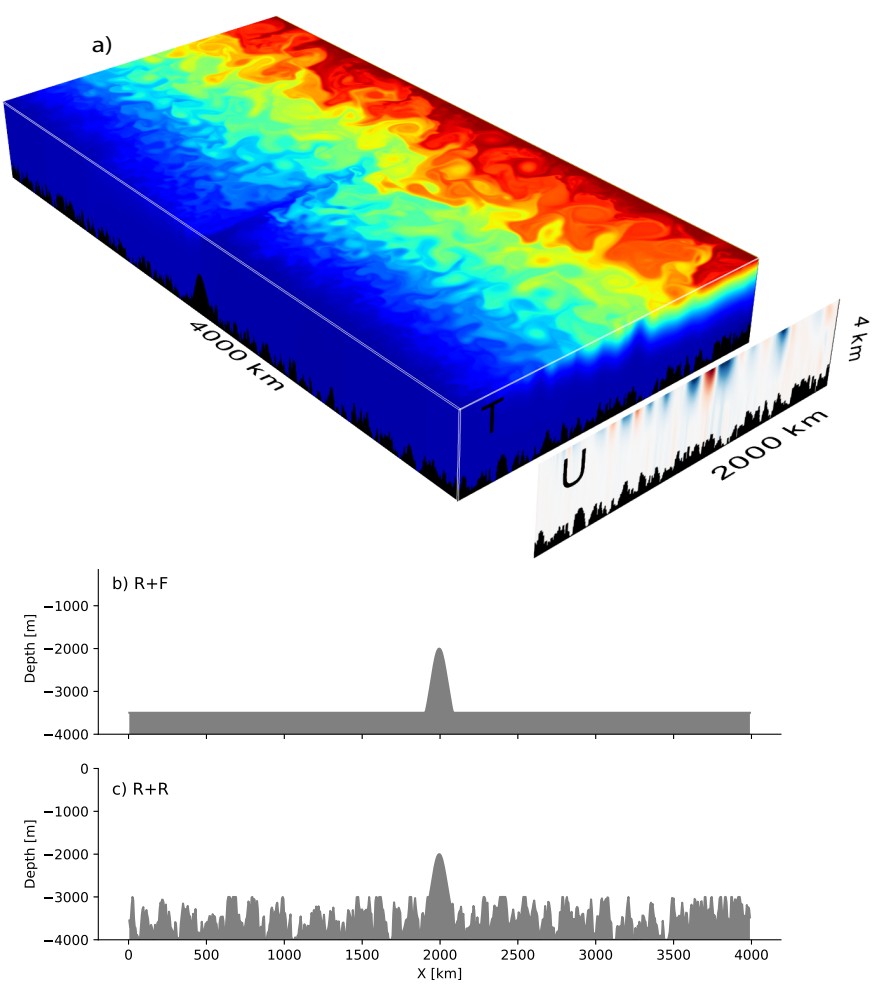

**Figure 2** 3D representation (a) of instantaneous temperature (rectangular box, color scale ranges from 0 to 8°C) and zonal velocity (vertical section) for the simulation R+R after 200 years. The domain is a 4000 km long - 2000 km wide reentrant channel. The maximum depth is 4000 m with irregular bottom topography bounded at 3000 m and a Gaussian-shaped ridge of 2000 m located at x=2000 km, which limits the ACC transport and generates a standing wave downstream as seen in the surface temperature. Topography at y=1000km in the two simulations: R+F (b) and R+R (c). The r.m.s height in R+R averaged within 50 x 50 km areas out of the region of the ridge is 250m.



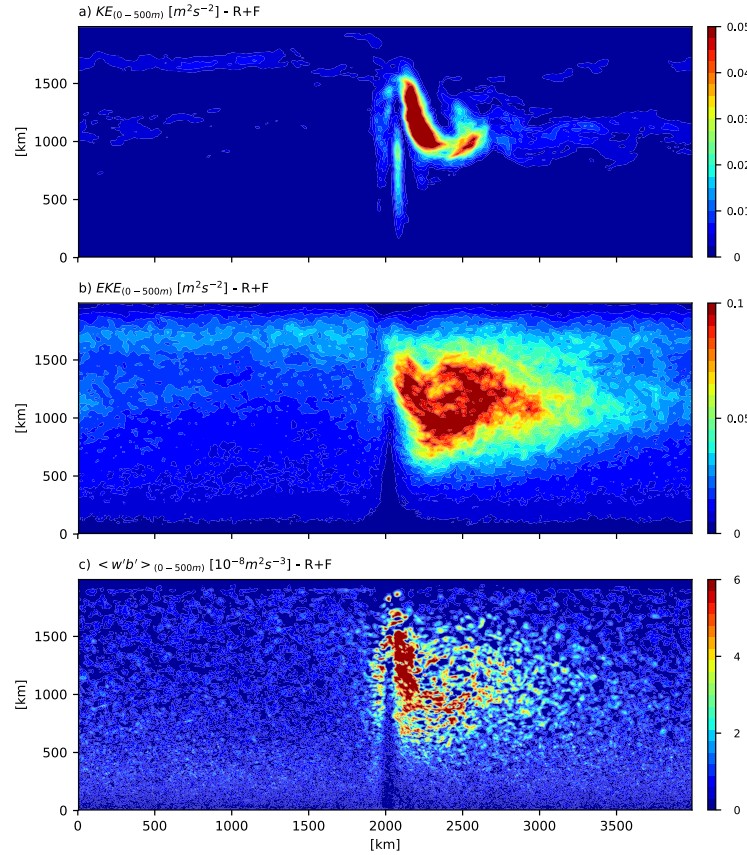

**Figure 3.** KE (m² s⁻²), EKE (m² s⁻²), $< \overline{w'b'} >$ (m² s⁻³) averaged between the surface and 500m. These fields have been
computed using instantaneous monthly data for the last ten years of the simulation R+F.





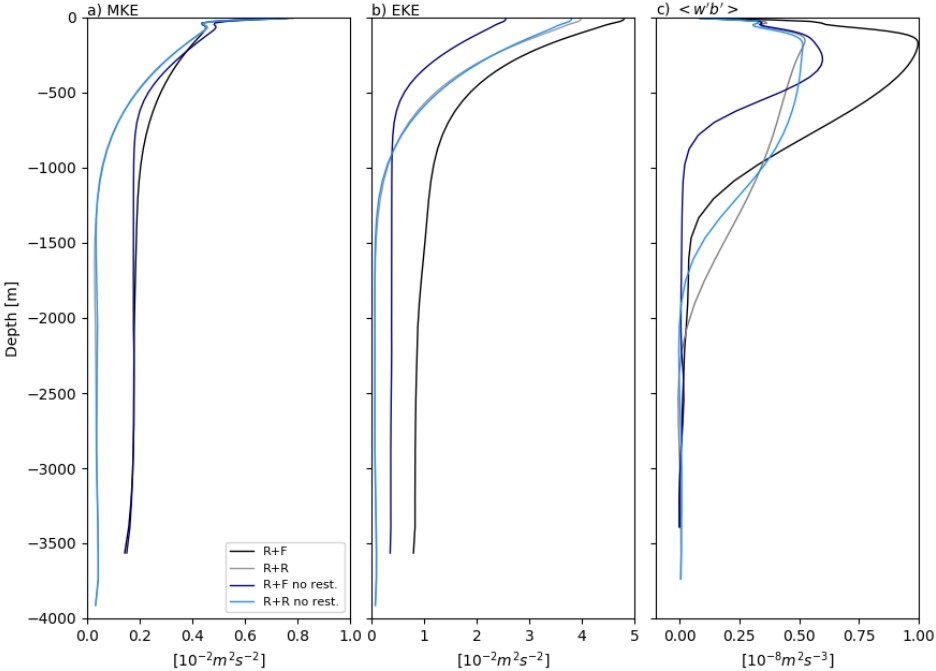

**Figure 4.** Vertical profile of (a) mean eddy kinetic energy (m$^2$ s$^{-2}$), (b) eddy kinetic energy (m$^2$ s$^{-2}$), and (c) $< \overline{w'b'} >$ the vertical eddy buoyancy flux (m$^2$ s$^{-3}$) as a signature of baroclinic energy transfer from eddy potential energy to eddy kinetic energy. Diagnostics use the last 10 years of the simulations and were averaged over the full domain in the zonal direction and between y=500km and y=1500km in the meridional direction. The velocity (u',v',w') and buoyancy (b') anomalies use to compute the eddy kinetic energy in (a) and the buoyancy flux in (c) are anomalies with respect to the 10-years temporal mean.


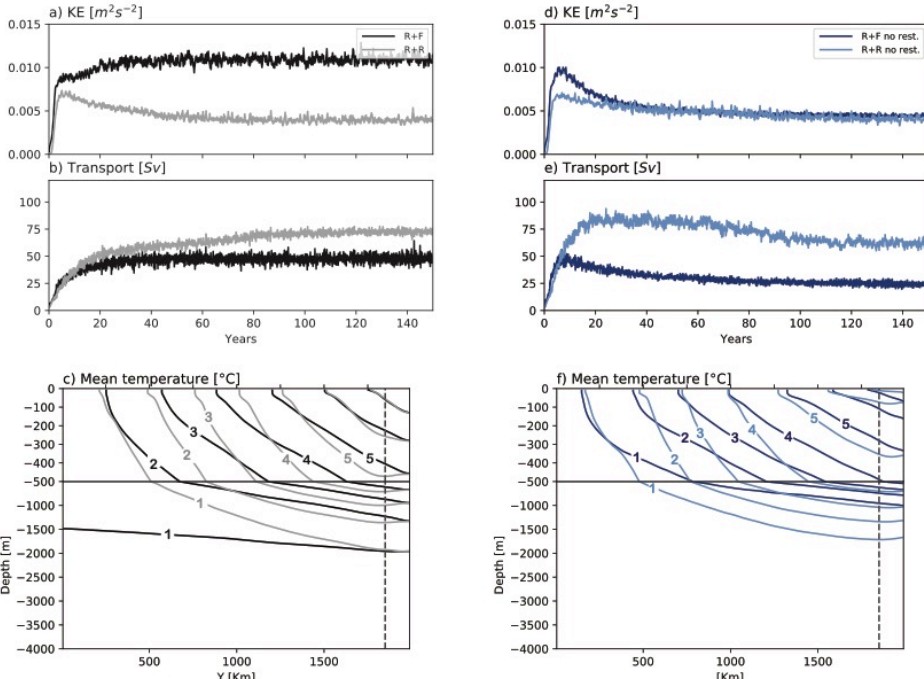

**Figure 5.** 150 years times series of domain integrated total kinetic energy (a,d; units m² s⁻²), transport (b,e; units Sv), and sections of zonally averaged mean temperature for the last ten years of the simulations (c,f; contours ranging from 1 to 7°C). Simulations with restoring at the northern boundary (R+R and R+F) are shown on the left and simulations without restoring (R+Rnr And R+Fnr) are shown on the right. In c), the vertical dashed line indicates the limit of the restoring zone at the northern boundary.





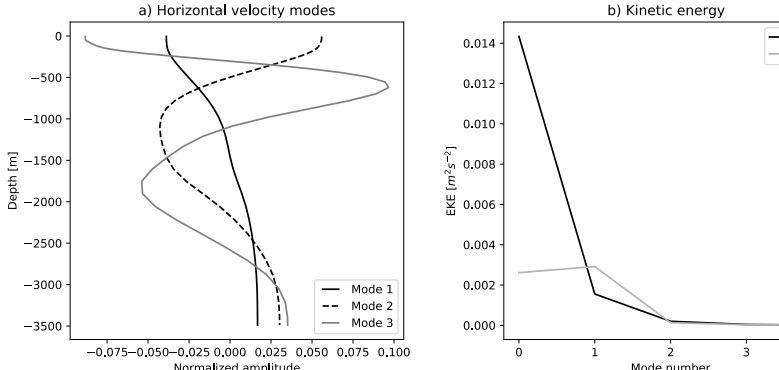

**Figure 6.** Normal mode analysis: a) the first three baroclinic modes at position [x=0; y=1000km] for the experiment flat, and b) kinetic energy ($m^2\ s^{-2}$) contained in each mode, with mode 0 corresponding to the barotropic mode. Normal mode analysis has been performed for profiles located at y=1000km and spaced by 100 km all along the zonal direction, and using monthly instantaneous outputs from the last ten years of simulations (R+F and R+R).





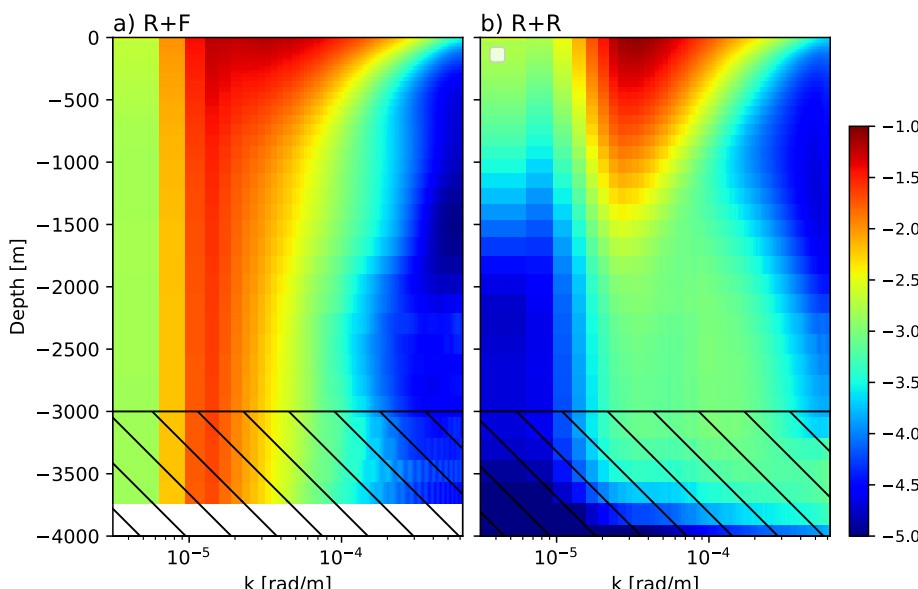

**Figure 7.** Kinetic energy power spectra (log10 m³ s⁻²) as a function of wavenumber (rad m⁻¹) and depth for simulations RF and RR. Spectra are built using instantaneous velocity taken each month of the last ten years of simulations.




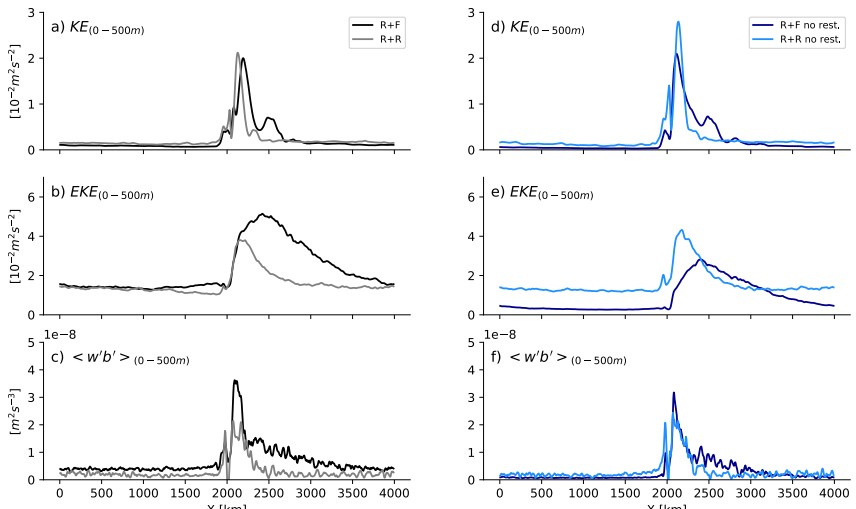

**Figure 8.** Zonal distribution of 10-years averaged (a,d) kinetic energy (m² s⁻²), (b,e) eddy kinetic energy (m² s⁻²), and (c,f) <w'b'> vertical eddy buoyancy flux (m² s⁻³). Quantities were averaged over the full domain in the meridional direction and between the surface and 500 m depth. The velocity (u',v',w') and buoyancy (b') anomalies use to compute the eddy kinetic energy in (a) and the buoyancy flux in (c,f) are anomalies with respect to the 10-years temporal mean.






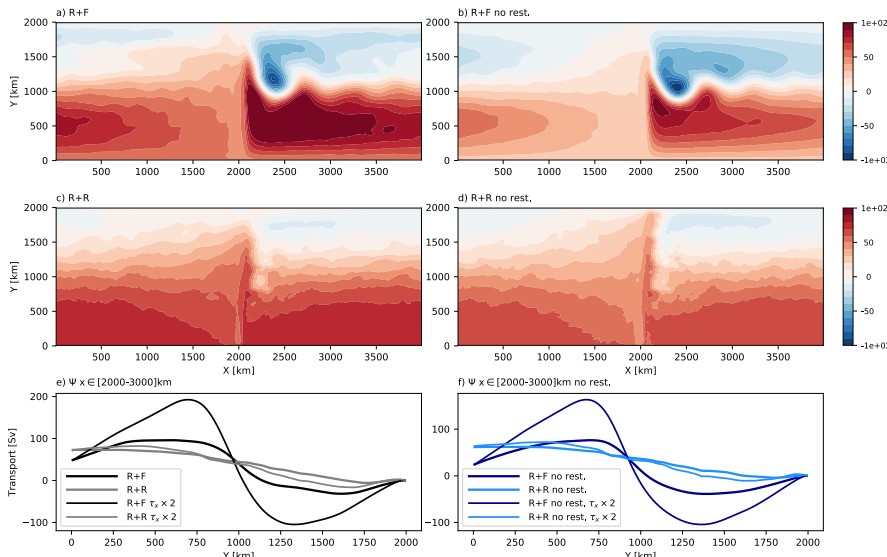

**Figure 9.** Barotropic streamfunction (Sv) for R+F (a), R+R (b), R+Fnr (c) and R+Rnr (d). Transport averaged between x=2000 and 3000 km for the set of reference simulation (bold lines) and simulations with maximum wind stress increased two fold, with restoring (e) and without restoring (f).





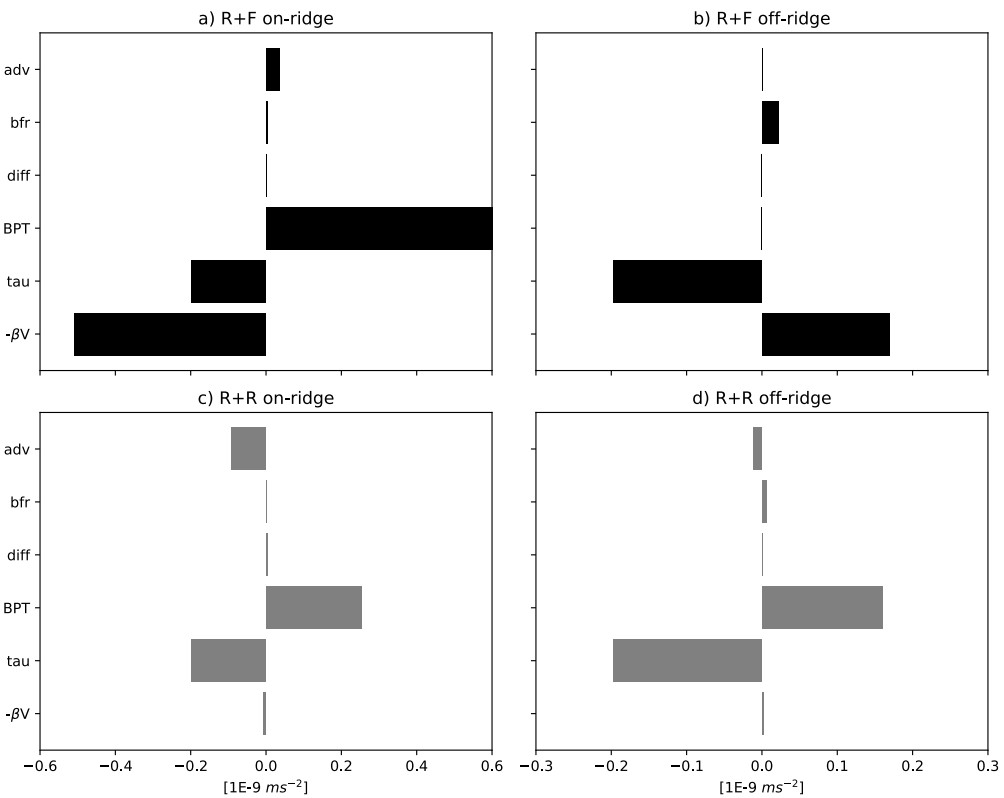

**Figure 10:** Depth integrated mean barotropic vorticity balance ($10^{-9}$ m s$^{-2}$) averaged between y=400 and y=600km for two zonal
portions of the domain: one under direct influence of the ridge (left column; between x=1500 and 2500km) and one including the
rest of the zonal domain (right column). The different terms are as follows: the advection of planetary vorticity (-βV), the wind
stress curl (tau), the bottom stress curl (bfr), the bottom pressure torque (BPT), the diffusion (which includes the effects of the
lateral diffusion and Asselin time filter) and the non-linear advection of vorticity (adv).





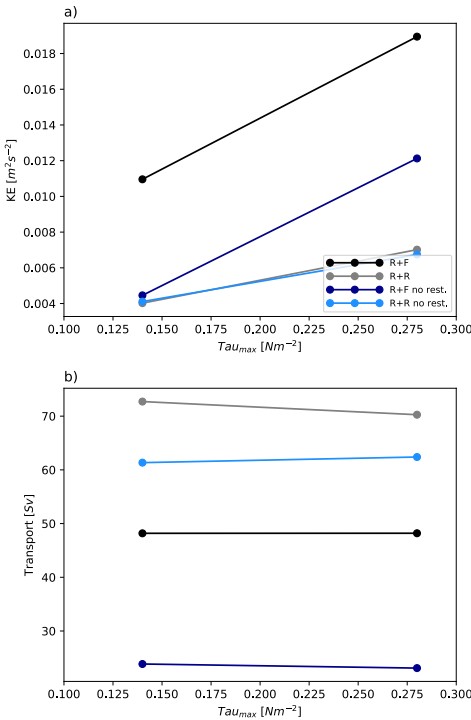


**Figure 11.** Sensitivity of (a) domain averaged KE (m² s⁻²), and zonal barotropic transport (Sv) to wind stress increase. Transport and KE values were averaged for the last ten years of simulation once equilibrium was achieved.





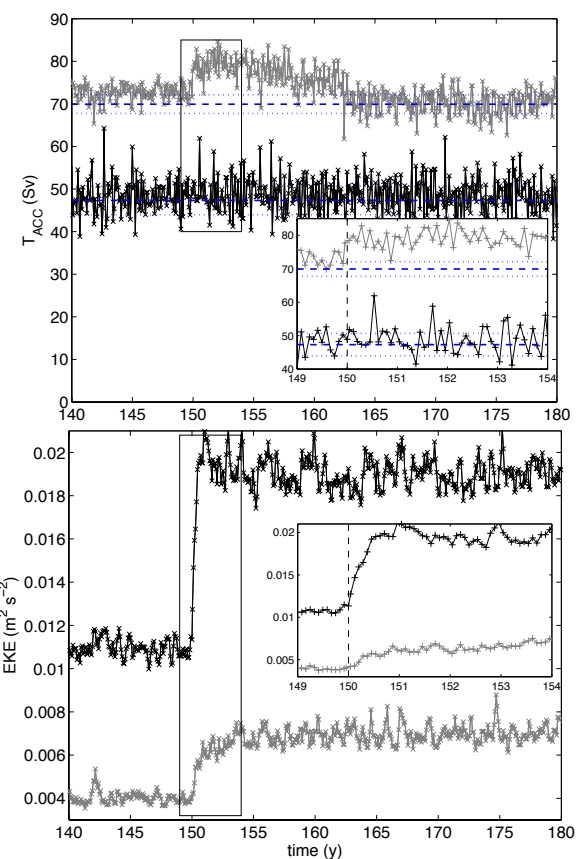


**Figure 12.** Times series of a) zonal transport [Sv] and b) total kinetic energy [$m^2$ $s^{-2}$] for simulations R+F (black lines) and (R+R) gray lines in response to an abrupt doubling of the wind stress at year 150. Inset provides details for the years 149-204.The blue dashed (resp. dotted) lines represent the average (resp. average +- 1standard deviation) transport between years 130 and 150.



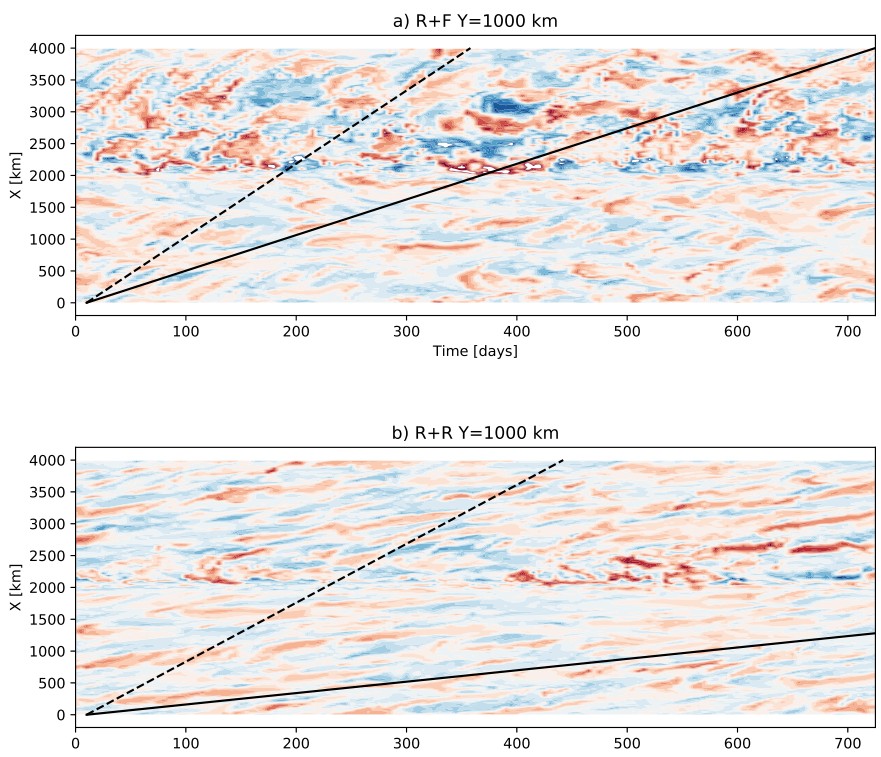

Figure 13. Hovmöller diagram of surface temperature anomalies at Y=1000 km as in Abernathey et al. [2013] built using 5-day averaged model outputs for a 2 years period. The dashed line indicates the zonal surface velocity and the continuous line indicates the barotropic velocity.