# Peer review of "Connecting flow-topography interactions, vorticity balance, baroclinic instability and transport in the Southern Ocean: the case of an idealized storm track"

_Ocean Science, 2020_

## Referee Comment (RC1) · Anonymous Referee #1 · 17 Mar 2020

The authors here try to assess the role of bathymetric roughness in establishing the mean circulation in the Southern Ocean. They do so using a series of idealised, zonally-reentrant simulations of primitive-equations on a beta plane.

The experiments performed and their analysis consist interesting numerical observations for how roughness affect the dynamical balances. However, the authors' attempt to explain the dynamical processes that take place and, thus, assess the dynamical role the bottom roughness brings about, are lacking. I have pointed out specific points below.

Overall, the paper is not very well-written and therefore **major revisions** are in place. Presentation is often sloppy and figures could definitely be improved. I find the numerical experiments performed here, as well as the accompanied analysis the authors went through, interesting and worthy of publication. However *not* at the manuscript's current form. Regarding dynamical explanation, e.g., section 4, I would like to see the arguments cleared up a bit; I provide specific comments below.

**Major points**

These need to be addressed by the authors.

1. general: Please number **all** equations.

2. line 50: Refrain from referring to a figure in a different paper! If the specific figure is crucial for the discussion then consider reproducing it here.

3. line 55, 59, ...: The authors use "*form stress*" and "*form drag*" interchangeably. Please choose one and stick to it throughout the manuscript. Personally I'd go with the former as this term does not always behave as drag (see Holloway's series of papers about the "Neptune effect").

4. line 88: This expression is completely different from that in Abernathey et al 2011. I believe (hope) this is a typo.

5. line 137: I would like to see a time-series of PE since, usually, that's what takes longer to equilibrate. It is important to see whether PE is equilibrated before one talks about time-mean isopycnal slopes.

6. paragraph starting line 185: The author's at this point try to explain why bottom roughness diminishes the gyres that can be found in the configuration with just the high-ridge. They compute the dominant terms in the Sverdrup balance (see figure 9 & 10). They do find that with and without roughness different terms dominate the Sverdrup balance. However, the paragraph here explains nothing! It's more like a chicken-egg argument. What the authors effectively say is that with roughness gyres turn off and the term $\beta V$ is not important. But of course, with no gyres term $\beta V$ can't be large. Do the authors try to argue here that roughness somehow implies that the vorticity balance **must** change from that in figure 10a to that in figure 10b and, therefore, the gyre turns off? If this is what they are trying to argue they need to back up the claim.

7. line 200: Regarding comparing the experiments with and without restoration at the northern boundary, the authors say: "*Most of our previous results are not qualitatively dependent on the choice of restoring the 200 northern stratification.*" However, from figure 4b,c I conclude the opposite. I see that experiments with 'nr' show opposite dependence on bottom roughness compared to the restoring experiments, especially in the upper 500m. Right?

8. line 206: If this is the total KE how come is smaller than EKE? I expect the total to be greater than any of its constituents.

9. line 228: I don't understand what are the "*general expectations drawn from eddy saturation theories*" the authors refer to at this point. Could they elaborate a bit? Also, citations should be relevant, potentially to the work by Straub *JPO* 1993, Marshall et al. *GRL* 2017, and Constantinou & Hogg *GRL* 2019.

10. line 250: "*As a consequence, only in the flat bottom configuration can the Sverdrup balance emerge.*": I don't understand what the authors want to say. In both R+F and R+R configurations the Sverdrup balance **balances** (see figure 10)! I guess they mean to write that when roughness is present, the balance is different

and diverges from the textbook picture that crucially involves the role barotropic Rossby waves? In either case, they should rephrase to make the text clearer.

11. line 255: "*... geophysical flows*": a citation to Rick Salmon is relevant here, e.g., "Baroclinic instability and geostrophic turbulence. *Geophys. Astrophys. Fluid Dyn.* **15**, 167-211 (1980)."

12. line 295-297: The authors here present baroclinic instability as the explanation for eddy saturation. But it has been established by a series of studies that bathymetry plays dominant role in eddy saturation (Thompson & Naveira Garabato *JPO* 2014, Katsumata *JPO* 2017, Barthel et al. *JPO* 2017, Youngs et al. *JPO* 2017, Constantinou and Hogg *GRL* 2019). The authors should update their explanation of eddy saturation.

13. figure 3: Add the same panels for the R+F experiment. Use the **same** colorscale.

14. figure 6: Caption mentions: "*Normal mode analysis has been performed for profiles located at $y = 1000km$ and spaced by $100km$ all along the zonal direction, and using monthly instantaneous outputs from the last ten years of simulations (R+F and R+R).*" I must admit that I don't understand what the authors are saying here. Please explain clearly or remove; I'd suggest the former.

15. figure 7a: This figure is puzzling since it shows that flow in R+F goes beyond $3500m$ in contrast with figure 2b. Also, what's the dashed region below $3000m$? Either remove or explain?

16. figure 9e+f: Please use different linestyles. The lines are barely distinguishable at the moment and it would be impossible for a colorblind reader.

**Minor comments/typos**

What follows is a list of suggestions. The authors can take them or leave them.

1. line 29: Hughes' name has a typo.

2. line 41: "*Further*" → "*In their setup, further*"

3. line 50: Refrain from referring to a figure in a different paper! If the specific figure is crucial for the discussion then consider reproducing it here.

4. line 57: "*periodic*" → "*zonally reentrant*"

5. line 66: Use subscripts in math, e.g., $L_x$, $L_y$.

6. line 74: Don't write, e.g., "$1.10^{-4}$"..., just write "$10^{-4}$". (Btw, why didn't you take $f_0 < 0$?)

7. line 83: $u_{10} = ...$ is erroneously repeated at the beginning of the line. Also I presume $u_0 = 10m\,s^{-1}$ should be $U_0 = ...$

8. line 84: Delete repeated "*formulation*". Also, why not writing the formulation for wind stress; it's just a single line equation?

9. line 117: Section 2.3 reads a bit weird at this point. Perhaps I'd suggest you discuss the vorticity balance further down when you are about to show the results of figure 10.

10. line 119: "*The **time-mean** BV equation...*"?

11. line 120: (a) $p_b$ needs a subscript; (b) use "·" and not "." for inner products; (c) refrain from putting parentheses around a single variable.

12. line 121: "$\beta$ the derivative of planetary vorticity" → I suggest defining this when it first appears further up.

13. line 121: "$V$ the integrated **time-mean** meridional vorticity"?

14. line 147: "*steady and turbulent*" → "*time-mean and transient*"?

15. line 154: "$1/(2 \cdot 10^{-4}m^{-1})$" is a pretty convoluted way to say "*5km*".

16. line 310: Nadeau & Ferrari (**2015**).

17. figure 5: The figure's quality is very poor. It only consist of lines, so the authors should be able to export it as a pdf/eps. Or, if they insist on using png/jpg, then I suggest they use higher dpi. Furthermore, please add a remark in the caption that the $z$-scale is not uniform. Also, consider reducing the $y$-limits of panels c) and f) down to only $-2500m$; there is nothing to be shown below that depth.

---

## Referee Comment (RC2) · Anonymous Referee #2 · 31 Mar 2020

This is an interesting paper that I enjoyed reading. The subject is of interest and relevant to an on-going discussion regarding Southern Ocean dynamics and circulation. There is a lot of material here covering, for example, energetics, modal decompositions, and spectral analysis, etc. I felt that this obfuscates the authors' message and means that they are left either trying to explain too much, or not explaining enough. The paper would be better served with a narrative that concentrates on the physical argument and only uses enough figures and analysis types to reinforce this. At the moment the paper's central argument is disguised by the extra material.

My main concerns are highlighted below, with additional minor comments appropriately titled.

1) The presence of gyres in a Southern Ocean model with f/h contours blocked by the northern boundary was not uncovered by Nadeau & Ferrari (2015). 'Highlighted' would be a better choice of word. There are numerous papers prior to, and contemporary with, Nadeau & Ferrari that show this same flow feature. Examples include Tansley & Marshall (2001) and Jackson et al. (2006), although there are plenty of others. A recent example that looks in detail at the formation of these gyres is Patmore et al. (2019).

2) At line 75 it is stated that 'no explicit diffusion' is used in the model. Later on diffusive terms are included in some of the figures, e.g. Figure 10. Does the model have diffusion? Or is it the case that vertical diffusive terms are included with no explicit horizontal diffusion? If this is the case, this means that the model is relying upon implicit diffusion in its advection scheme, which may have implications for the form of its overturning. Is the model's residual overturning quasi-adiabatic, as achieved in Abernathey et al. (2011)?

3) At lines 167-169 the authors write that 'rough topography limits the eddy energy at the location of the stationary meanders but also favors the persistence of the eddy energy far from the ridge.' On my first read through this felt like an important point. However, I don't think it's particularly born out by Figure 8. Rather, Figure 8 shows the more local confinement to the ridge due to the rough bottom. It's the flat bottom experiments that truly favour downstream persistence of EKE.

4) At lines 185-188 the authors briefly discuss the PV budget in experiment R+F. I expected something to be said about the much larger contribution of nonlinear vorticity advection over the ridge for this experiment. The import/export of vorticity into/out of this area could be an important reflection of the change in dynamics.

5) Section 3.3 discusses the experiments in which no restoring takes place at the northern boundary. Something that gets overlooked here, but is apparent in Figure 5, is that the isotherms in both experiments with bottom roughness are at similar depths on the northern boundary. This suggests to me that the rough bathymetry may be constraining the transport by allowing geostrophic return flow at greater depths than the ridge alone. This would allow for deeper isotherms and higher circumpolar transport. Calculating the average depth of an isotherm, instead of zonally averaging across temperature classes, would make this clearer.

6) Section 4.1 is where I found myself becoming unstuck. This discussion is very important to the message of the paper and starts well by laying out a series of 5 points that the authors aim to connect. I think the progression of the discussion would be well-served by using these points as headers to the relevant paragraphs. This would help signpost the path for the reader. It might also be useful to introduce some of this material earlier in order to ease the cognitive burden at this point in the paper. Currently, this section is confusingly written and I found myself becoming confused by the authors' argument. Section 4.2 makes some very interesting and important points. It connects the authors' discussion with other relevant literature. However, I think this could also do with being revised in order to ensure it is as clear as possible.

Minor Comments

line 24 : 'The real SO. . .seems to lie in the "rough bottom/no wind-driven gyre" regime.' The SO clearly does have gyres and ending the abstract on something that causes less confusion would be better.

line 30 : Scotian -> Scotia

line 33 :' referred to' at end of line.

line 45 : 'thought to dissipate'

lines 44-56 : These two paragraphs feel disconnected from the rest of the introduction; the switch from discussing gyre dynamics to bathymetry is very abrupt.

line 87 : Equation immediately following. I think there's a typo in the form of the vertical restoring. This is very similar to the cited example of Abernathey et al. (2011), but not quite the same.

line 90 : A restoring of 7 /day would be very strong, is it 1/(7 days)?

line 140 : It would be very helpful for the reader to also specify the mean transports of the currents.

line 182 : Strictly speaking it isn't that the 'pressure torque is no more effective', its that the pressure torque is zero.

line 214 : 'In the presence'

References

Abernathey, R., J. Marshall, and D. Ferreira, 2011: The dependence of Southern Ocean meridional overturning on wind stress. J. Phys. Oceanogr., 41, 2261–2278.

Jackson, L., C. W. Hughes, and R. G. Williams, 2006: Topographic control of basin and channel flows: The role of bottom pressure torques and friction. J. Phys. Oceanogr., 36, 1786–1805.

Patmore, R. D., P. R. Holland, D. R. Munday, A. C. Naveira Garabato, D. P. Stevens, and M. P. Meredith, 2019: Topographic control of Southern Ocean gyres and the Antarctic Circumpolar Current: A barotropic perspective. J. Phys. Oceanogr., 49, 3221–3244, doi:10.1175/JPO–D–19–0083.1.

Tansley, C. E. and D. P. Marshall, 2001: On the dynamics of wind-driven circumpolar currents. J. Phys. Oceanogr., 31, 3258–3273.

---

## Author Comment (AC1) · 29 Jun 2020

**Reply to comments by reviewer #1 on "Connecting flow-topography interactions, vorticity balance, baroclinic instability and transport in the Southern Ocean: the case of an idealized storm track" by Julien Jouanno and Xavier Capet, July 2020**

We thank the reviewers for their thoughtful comments and have done our best to address them. Before we proceed to the specific responses, we wish to highlight a general aspect of our review work. Both reviewers seem to have had problems with the sequential structure of the manuscript (broadly section 3 is a description of results that do not seem particularly connected; sections 4 provides interpretations/discussions to attempt to connect/unify the pieces together). We understand that this organization is less common in our field than in other ones. We have attempted to reorganize the manuscript differently but we ended up not doing so, mainly for the reason that the chains of processes we propose in section 4 is complex and is more clearly explained once all supporting material it needs has been presented. This being said, we have carefully rewritten some key parts of the manuscript (including the final paragraph of the introduction) to make sure the reader is well aware that the key interpretations will come in section 4. Therefore, and with the improvements in the lay out of the discussion in 4.1 (following the suggestion of reviewer 2), a reader like reviewer 2 who would feel "stuck" in section 3, could more naturally jump to section 4 for a scanning of our interpretations.

The authors here try to assess the role of bathymetric roughness in establishing the mean circulation in the Southern Ocean. They do so using a series of idealized, zonally-reentrant simulations of primitive-equations on a beta plane.

The experiments performed and their analysis consist interesting numerical observations for how roughness affect the dynamical balances. However, the authors' attempt to explain the dynamical processes that take place and, thus, assess the dynamical role the bottom roughness brings about, are lacking. I have pointed out specific points below.

Overall, the paper is not very well-written and therefore major revisions are in place. Presentation is often sloppy and figures could definitely be improved. I find the numerical experiments performed here, as well as the accompanied analysis the authors went through, interesting and worthy of publication. However *not* at the manuscript's current form. Regarding dynamical explanation, e.g., section 4, I would like to see the arguments cleared up a bit; I provide specific comments below.

We thank the Reviewer for his valuable input and refer to detailed responses to all of its comments below. We have tried to improve the text and figures in many places, in particular with the aim to make our dynamical interpretations as clear as possible.

**Major points**

These need to be addressed by the authors.

1. general: Please number all equations.

All equations have been numbered.

2. line 50: Refrain from referring to a figure in a different paper! If the specific figure is crucial for the discussion then consider reproducing it here.

We remove the reference to the figure, here and after, and now we only retain the reference to the paper Goff and Arbic 2010.

3. line 55, 59, ...: The authors use "*form stress*" and "*form drag*" interchangeably. Please choose one and stick to it throughout the manuscript. Personally I'd go with the former as this term does not always behave as drag (see Holloway's series of papers about the "Neptune effect").

Thanks. Following your suggestion, we stick with form stress all along the manuscript.

4. line 88: This expression is completely different from that in Abernathey et al 2011. I believe (hope) this is a typo.

Thanks, yes this was a typo. It has been corrected.

Line 137: I would like to see a time-series of PE since, usually, that's what takes longer to equilibrate. It is important to see whether PE is equilibrated before one talks about time-mean isopycnal slopes.

Indeed the reviewer is right about the fact that PE is not fully equilibrated after 150 years as illustrated in Figure R1 below. We have extended the simulations for another 100 years. Simulations are closer to PE equilibrations after 250 years of simulation although a slight adjustment is still visible for simulations without northern restoring (Figure R1). This being said, comparison of the density fields around t=150 y and t=250 y indicates that adjustments are minor and in particular that stratification differences between the sensitivity runs are large compared to stratification drifts diagnosed for each model run (see Figure R2). Therefore, we have stuck with our original analysis period year 140-150.

Figure R1. Time evolution of PE in the four simulations.

Figure R2. Time mean isotherms averaged over the years 140 to 150 (continuous line) and year 240 to 250 (dashed line). Color code is the same as in Figures 5c,f of the manuscript.

5. line 185: The author's at this point try to explain why bottom roughness diminishes the gyres that can be found in the configuration with just the high-ridge. They compute the dominant terms in the Sverdrup balance (see figure 9 & 10). They do find that with and without roughness different terms dominate the Sverdrup balance. However, the paragraph here explains nothing! It's more like a chicken-egg argument. What the authors effectively say is that with roughness gyres turn off and the term  $\beta V$  is not important. But of course, with no gyres term  $\beta V$  can't be large. Do the authors try to argue here that roughness somehow implies that the vorticity balance must change from that in figure 10a to that in figure 10b and, therefore, the gyre turns off? If this is what they are trying to argue they need to back up the claim.

Although we agree with the reviewer that the chicken and egg trap shall be avoided, we believe the text clearly sticks to a descriptive objective at this place and does not try to propose any interpretations. Interpretations on the BV balance are presented in section 4.1 and have been carefully rewritten to make sure there is no chicken and no egg there either. Precisely, we will only be making the point that R+F has no choice but to balance the wind curl input with  $\beta$ V outside the ridge area while R+R does have more freedom and in fact balances wind curl input with bottom pressure torque.

6. line 200: Regarding comparing the experiments with and without restoration at the northern boundary, the authors say: "*Most of our previous results are not qualitatively dependent on the choice of restoring the northern stratification*." However, from figure 4b,c I conclude the opposite. I see that experiments with 'nr' show opposite dependence on bottom roughness compared to the restoring experiments, especially in the upper 500m. Right?

You are right and this was discussed in same paragraph, but somehow embedded in the discussion of total KE sensitivity. We now made this discussion on EKE sensitivity in Figures 4b,c more explicit :

"Second, bottom roughness strongly decreases total KE when restoring is applied while total KE is very weakly affected when no restoring is applied (Figure 5a,d). We attribute this to the fact that the more efficient release of available potential energy in the absence of rough bathymetry (Figure 4c), that lead to larger EKE in the upper 500 m (Figure 4b), can significantly modify the ACC thermohaline structure in the simulations without restoring whereas it cannot when tightly constrained by the restoring (compare the departures between isotherms in Figs.5c and f). Further elaboration is provided in Section 4."

**And in Section 4 :**

"The reduced baroclinicity and zonal transport in R+F and R+Fnr can thus be seen as the manifestation of the boundary current effect on local baroclinic instability in the lee of the ridge. In the simulations without restoring this manifestation on baroclinic instability is less evident because the mean thermohaline structure of the ACC has significantly more freedom to adjust in response to the strength of baroclinic instability processes. In turn, this response of the mean state lead to a negative feedback by modulating the intensity of baroclinic processes which ends up being quite similar with and without rough bathymetry in the absence of northern restoring (compare EKE and APE release rate for R+Fnr and R+Rnr in Figures 4c and 8f)."

7. line 206: If this is the total KE how come is smaller than EKE? I expect the total to be greater than any of its constituents.

Here we referred to Figure 5 where we show total KE. For clarity, we add reference to the figure. Moreover, we add in Figure 4 caption more details on how we compute the "MKE" (kinetic energy of the mean flow) so there should be no more ambiguity: "The kinetic energy of the mean flow (MKE, a) is computed using 10-years averaged velocities." On the other hand, if you remark concerned Fig. 8 (which we did not refer to at line 206), please note that there was an error in the legends/captions of that figure and that panels a) and d) show the kinetic energy of the mean flow.

8. line 228: I don't understand what are the "*general expectations drawn from eddy saturation theories*" the authors refer to at this point. Could they elaborate a bit? Also, citations should be relevant, potentially to the work by Straub *JPO* 1993, Marshall et al. *GRL* 2017, and Constantinou & Hogg *GRL* 2019.

We agree the sentence was vague. We preferred to remove it since discussion on the eddy saturation process in given in section 4.2.

9. line 250: "*As a consequence, only in the flat bottom configuration can the Sverdrup balance emerge.*": I don't understand what the authors want to say. In both R+F and R+R configurations the Sverdrup balance balances (see figure 10)! I guess they mean to write that when roughness is present, the balance is different and diverges from the textbook picture that crucially involves the role barotropic Rossby waves? In either case, they should rephrase to make the text clearer.

Sverdrup balance specifically refers to the dominant balance between the wind stress curl and planetary vorticity term ( $\beta$ V) as classically understood. Such balance is a good approximation of the vorticity balance in the "R+F" case (Figure 10b) but not in R+R. We're not trying to say more than that. The text has been rewritten in such a way that, we think, no confusion can happen.

Line 255: "... *geophysical flows*": a citation to Rick Salmon is relevant here, e.g., "Baroclinic instability and geostrophic turbulence. *Geophys. Astrophys. Fluid Dyn.* 15, 167-211 (1980)."

We agree and choose to refer to this study: Salmon, R., Holloway, G., & Hendershott, M. C. (1976). The equilibrium statistical mechanics of simple quasi-geostrophic models. Journal of Fluid Mechanics, 75(4), 691-703.

10. line 295-297: The authors here present baroclinic instability as the explanation for eddy saturation. But it has been established by a series of studies that bathymetry plays dominant role in eddy saturation (Thompson & Naveira Garabato JPO 2014, Katsumata JPO 2017, Barthel et al. JPO 2017, Youngs et al. JPO 2017, Constantinou and Hogg GRL 2019). The authors should update their explanation of eddy saturation.

Thanks for these references we missed. We complete this section as follows:

"Baroclinic instability, which is the main source of energy for the mesoscale eddy field in the SO consumes the APE imparted by wind-driven upwelling. It occurs in such a way that additional energy input by the wind enhances EKE but leaves APE and ACC transport nearly unchanged. This contributes to the so-called eddy saturation effect which limits the sensitivity of the circumpolar transport to changes in the wind forcing magnitude (Morrison and Hogg 2012, Munday et al. 2013, Marshall et al, 2017). Processes involving the barotropic circulation and its interaction with the bathymetry may also participate to reduce the sensitivity of the ACC's baroclinicity. Specifically, the standing meanders that forms through the interaction of the barotropic flow with the topography contribute to the bottom form stress and may also participate to the saturation process (Thompson & Naveira Garabato, 2014, Katsumata, 2017). Constantinou and Hogg (2019) recently highlight the role played by the eddy production through lateral shear instabilities of the barotropic circulation or interaction of the barotropic current with the topography, in establishing the eddy saturated state of the Southern Ocean. Overall, our findings confirm the robustness of the saturation process with respect to major changes in model configuration, which translate into varied baroclinic instability regimes/efficiency (as previously noted in Nadeau et al. 2013), but also flow with varied barotropic dynamics and a wide range of ACC transports."

11. figure 3: Add the same panels for the R+F experiment. Use the same colorscale.

Same panels for R+R have been added in Figure 3 and the same colorbar is used.

12. figure 6: Caption mentions: "Normal mode analysis has been performed for profiles located at y = 1000km and spaced by 100km all along the zonal direction, and using monthly instantaneous outputs from the last ten years of simulations (R+F and R+R)." I must admit that I don't understand what the authors are saying here. Please explain clearly or remove; I'd suggest the former.

The caption has been edited as follows : "The kinetic energy given in (b) results from a combination of : spatial averaging over 40 profiles taken at the central latitude (y=1000 km) and regularly spaced in longitude all along the channel; and temporal averaging 120 snapshots obtained at monthly frequency over the last ten years of simulations R+F and R+R."

13. figure 7a: This figure is puzzling since it shows that flow in R+F goes beyond 3500m in contrast with figure 2b. Also, what's the dashed region below 3000m? Either remove or explain?

This was to indicate the depths for which the spectrum was "polluted" by the rough topography. We modified the figure so we now only consider the depths entirely filled by the ocean outside the ridge (i.e. 3000 m in R+R and 3500 in R+F). The message to be taken from the figure remains unchanged.

14. figure 9e+f: Please use different linestyles. The lines are barely distinguishable at the moment and it would be impossible for a colorblind reader.

We now show model runs with increased wind stress with dashed lines. Lines are much more distinguishable. Thanks for reminding us about this.

**Minor comments/typos**

What follows is a list of suggestions. The authors can take them or leave them.

1. line 29: Hughes' name has a typo.

**Corrected.**

2. line 41: "Further"  $\rightarrow$  "In their setup, further"

Thanks, we add this sentence.

3. line 50: Refrain from referring to a figure in a different paper! If the specific figure is crucial for the discussion then consider reproducing it here.

Reference to the figure has been removed.

4. line 57: "periodic"  $\rightarrow$  "zonally reentrant"

**Corrected**

5. line 66: Use subscripts in math, e.g.,  $L_X$ ,  $L_V$ .

Corrected here and elsewhere.

6. line 74: Don't write, e.g., "1.10-4"..., just write "10-4". (Btw, why didn't you take  $f_0 < 0$ ?)

Corrected. An indeed, f0 is negative, so we have corrected the value

7. line 83:  $u_{10} = ...$  is erroneously repeated at the beginning of the line. Also I presume  $u_0 = 10 \text{ m s}^{-1}$  should be  $U_0 = ...$

Thanks, corrected.

8. line 84: Delete repeated "*formulation*". Also, why not writing the formulation for wind stress; it's just a single line equation?

Corrected. The Large and Yeager equation is not a single line expression, since it includes polynomial expressions for the drag coefficient, so we prefer not to write the wind stress formulation.

9. line 117: Section 2.3 reads a bit weird at this point. Perhaps I'd suggest you discuss the vorticity balance further down when you are about to show the results of figure 10.

We are not comfortable in introducing the vorticity balance in Section 3.2C so we would prefer to let it in the methodology section. We have modified the text and hope it reads less weird now.

10. line 119: "The time-mean BV equation..."?

Correct. We rephrased.

11. line 120: (a) pb needs a subscript; (b) use "." and not "." For inner products; (c) refrain from putting parentheses around a single variable. C5

This has been corrected.

12. line 121: " $\beta$  the derivative of planetary vorticity"  $\rightarrow$  I suggest defining this when it first appears further up.

This has modified. Thanks.

13. line 121: "V the integrated time-mean meridional vorticity"?

Corrected

14. line 147: "steady and turbulent"  $\rightarrow$  "time-mean and transient"?

Corrected.

-4 -115. line 154: "1/(2 · 10 m)" is a pretty convoluted way to say "5km".

Corrected

16. line 310: Nadeau & Ferrari (2015).

Corrected thanks.

17. figure 5: The figure's quality is very poor. It only consists of lines, so the authors should be able to export it as a pdf/eps. Or, if they insist on using png/jpg, then I suggest they use higher dpi. Furthermore, please add a remark in the caption that the z-scale is not uniform. Also, consider reducing the y-limits of panels c) and f) down to only -2500m; there is nothing to be shown below that depth.

As mentioned above, figures were saved in pdf, with high quality. But their inclusion in the word documents degraded their quality. We will take good care, if the manuscript is accepted, that published version will respect the high quality of our figures.

---

## Author Comment (AC2) · 29 Jun 2020

**Reply to comments by reviewer #2 on "Connecting flow-topography interactions, vorticity balance, baroclinic instability and transport in the Southern Ocean: the case of an idealized storm track" by Julien Jouanno and Xavier Capet, July 2020**

We thank the reviewers for their thoughtful comments and have done our best to address them. Before we proceed to the specific responses, we wish to highlight a general aspect of our review work. Both reviewers seem to have had problems with the sequential structure of the manuscript (broadly section 3 is a description of results that do not seem particularly connected; sections 4 provides interpretations/discussions to attempt to connect/unify the pieces together). We understand that this organization is less common in our field than in other ones. We have attempted to reorganize the manuscript differently but we ended up not doing so, mainly for the reason that the chains of processes we propose in section 4 is complex and is more clearly explained once all supporting material it needs has been presented.  This being said, we have carefully rewritten some key parts of the manuscript (including the final paragraph of the introduction) to make sure the reader is well aware that the key interpretations will come in section 4. Therefore, and with the improvements in the lay out of the discussion in 4.1 (following the suggestion of reviewer 2), a reader like reviewer 2 who would feel "stuck" in section 3, could more naturally jump to section 4 for a scanning of our interpretations.

This is an interesting paper that I enjoyed reading. The subject is of interest and relevant to an on-going discussion regarding Southern Ocean dynamics and circulation. There is a lot of material here covering, for example, energetics, modal decompositions, and spectral analysis, etc. I felt that this obfuscates the authors' message and means that they are left either trying to explain too much, or not explaining enough. The paper would be better served with a narrative that concentrates on the physical argument and only uses enough figures and analysis types to reinforce this. At the moment the paper's central argument is disguised by the extra material.

We are very pleased the Reviewer enjoyed reading the paper and found it relevant. We thank her/him for the time taken to review our manuscript. We discuss each of the comments sequentially below. We have seriously contemplated the possibility to narrow down the scope of the paper but we remained, in the end, quite convinced that all the pieces of the story are important. We have tried to improve several bits of text with the intent to make this more apparent. The only part that could possibly be removed, we think, is the final comment on local versus global instability because it does not seem relevant to rationalize the relative behaviours of our various ACCs. On the other hand, we feel that it can be important to keep it because it is such a key element of Abernathey and Cessi (2014) and may otherwise seem strangely absent.

My main concerns are highlighted below, with additional minor comments appropriately titled.

1) The presence of gyres in a Southern Ocean model with f/h contours blocked by the northern boundary was not uncovered by Nadeau & Ferrari (2015). 'Highlighted' would be a better choice of word. There are numerous papers prior to, and contemporary with, Nadeau & Ferrari that show this same flow feature. Examples include Tansley & Marshall (2001) and

Jackson et al. (2006), although there are plenty of others. A recent example that looks in detail at the formation of these gyres is Patmore et al. (2019).

Thanks for this suggestion and for the references. The section has been rephrased as follows:

"Another potentially important aspect of the dynamics through which ridges affect the SO circulation is the formation of closed recirculating gyres driven by Sverdrup like dynamics that co-exist with the circumpolar flow (Tansley & Marshall 2001, Jackson et al. 2006). From idealized numerical simulations of the ACC, it was recently highlighted by Nadeau and Ferrari (2015) that increasing wind intensity leads to increasing gyre circulation without modification of the circumpolar transport, suggesting that the saturation of the circumpolar transport with increasing winds may be connected with gyre dynamics. Patmore et al. (2019) further highlight that ridge geometry is important for determining gyre strength and the net zonal volume transport."

2) At line 75 it is stated that 'no explicit diffusion' is used in the model. Later on diffusive terms are included in some of the figures, e.g. Figure 10. Does the model have diffusion? Or is it the case that vertical diffusive terms are included with no explicit horizontal diffusion? If this is the case, this means that the model is relying upon implicit diffusion in its advection scheme, which may have implications for the form of its overturning. Is the model's residual overturning quasi-adiabatic, as achieved in Abernathey et al. (2011)?

Yes the horizontal advection scheme has some implicit diffusion. Its contribution to the energy balance in a similar channel configuration has been described with details in Jouanno et al. (2016). The model also includes some vertical diffusion : *"The vertical diffusion coefficients are given by a Generic Length Scale (GLS) scheme with a k-ε turbulent closure (Reffray et al. 2015)."*. But as shown below, the residual overturning computed with the last ten years of the R+R simulations is quasi adiabatic in the interior. We have added a parenthesis saying "(implicit diffusion can be diagnosed whenever necessary; see for instance Jouanno et al, 2016)".

[Figure]

*Figure R2: residual overturning in R+R. Diapycnal transformations occurs at the northern boundary where restoring is active, in the upper ocean, and at the southern boundary where the mixed-layer can reach the bottom.*

3) At lines 167-169 the authors write that 'rough topography limits the eddy energy at the location of the stationary meanders but also favors the persistence of the eddy energy far from the ridge.' On my first read through this felt like an important point. However, I don't think it's particularly born out by Figure 8. Rather, Figure 8 shows the more local confinement to the ridge due to the rough bottom. It's the flat bottom experiments that truly favour downstream persistence of EKE.

Indeed, this is somehow hidden by the meridional averaging in Figure 8. We now show spatial map of EKE in Figure 3 that illustrate how "rough topography limits the eddy energy at the location of the stationary meanders but also slightly increases EKE downward over a restricted latitude band. This being said, this is not something that we make sense of later on or is used in our interpretations. Therefore, we have removed this bit.

4) At lines 185-188 the authors briefly discuss the PV budget in experiment R+F. I expected something to be said about the much larger contribution of nonlinear vorticity advection over the ridge for this experiment. The import/export of vorticity into/out of this area could be an important reflection of the change in dynamics.

Indeed, on the ridge there is a change of sign of the non-linear vorticity advection between R+F and R+R we're having difficulty interpreting. Since the contribution of this term to the barotropic vorticity balance is second-order and we attempt to keep the focus of the study on first-order sensitivities we decided not to mention this in the revision.

5) Section 3.3 discusses the experiments in which no restoring takes place at the northern boundary. Something that gets overlooked here, but is apparent in Figure 5, is that the isotherms in both experiments with bottom roughness are at similar depths on the northern

boundary. This suggests to me that the rough bathymetry may be constraining the transport by allowing geostrophic return flow at greater depths than the ridge alone. This would allow for deeper isotherms and higher circumpolar transport. Calculating the average depth of an isotherm, instead of zonally averaging across temperature classes, would make this clearer.

We provide Fig. R3 to answer this remark. In Fig. R3 we show the mean depth of isotherm in all four simulations. It helps figure out the similarities between simulations in terms of stratification. In fact, temperature fields at the northern boundary do not appear to be more similar in the 2 simulations with rough bottom than they are in the 2 simulations with smooth bottom. The reviewer may be right that a complex interplay between the stratification that gets established in the simulations and the associated transverse circulation also contributes to explaining some of the transport variations. In fact we allude to this in relation to the long or secondary adjustments in transport that are seen in the rough simulations (Fig. 5 and 12). This being said and despite some efforts (prior to submission and during the review) we have not been able to identify processes at play that would help the reader make sense of these second-order aspects. Given the relative complexity of the message on dominant processes at play we would prefer not add details on the stratification/transverse flow subject.

[Figure]

*Figure R3. Field of zonal and time averaged isothermal depth for all 4 simulations.*

6) Section 4.1 is where I found myself becoming unstuck. This discussion is very important to the message of the paper and starts well by laying out a series of 5 points that the authors aim to connect. I think the progression of the discussion would be well-served by using these points as headers to the relevant paragraphs. This would help signpost the path for the reader. It might also be useful to introduce some of this material earlier in order to ease the cognitive burden at this point in the paper. Currently, this section is confusingly written and I found myself becoming confused by the authors' argument. Section 4.2 makes some very interesting and important points. It connects the authors' discussion with other relevant literature. However, I think this could also do with being revised in order to ensure it is as clear as possible.

We agree, this discussion is quite dense. The different processes involved in the ACC are so interconnected that we think they need to be presented together. We have reworded some parts of this discussion and, most importantly, we have introduced headers as suggested by reviewer 2. We like it a lot and we think this will be helpful to readers.

Minor Comments

line 24 : 'The real SO. . .seems to lie in the "rough bottom/no wind-driven gyre" regime.' The SO clearly does have gyres and ending the abstract on something that causes less confusion would be better.

We have reworded that last sentence so as to cause less confusion: "The real SO having both gyres and ACC saturation time scales typical of our "no gyre" simulations may be in an intermediate regime in which mesoscale topography away from major ridges provides partial and localized support for bottom form stress/pressure torque."

line 30 : Scotian -> Scotia

Thanks. Corrected.

line 33 :' referred to' at end of line.

Thanks. Corrected.

line 45 : 'thought to dissipate'

Corrected here and one another location.

lines 44-56 : These two paragraphs feel disconnected from the rest of the introduction; the switch from discussing gyre dynamics to bathymetry is very abrupt.

We find the transition rather natural since "ridges" are bathymetric features. In the previous paragraph we discussed the known impact of the ridge on the circulation, and in these two paragraphs we discuss the known impact of smaller topographic features. In fact, to make sure the connection is clear to the reader, we have reworded the second paragraph which has been shortened and merged with the first one. We hope this will convince reviewer 2.

line 87 : Equation immediately following. I think there's a typo in the form of the vertical restoring. This is very similar to the cited example of Abernathey et al. (2011), but not quite the same.

Thanks, there was a typo that has been corrected.

line 90 : A restoring of 7 /day would be very strong, is it 1/(7 days)?

We use a restoring time scale of 7 days. To avoid confusion this is described as follows : "The relaxation coefficient varies linearly from 0 at y=1900 km to 7 days at Ly."

line 140 : It would be very helpful for the reader to also specify the mean transports of the currents.

Transport have been added.

line 182 : Strictly speaking it isn't that the 'pressure torque is no more effective', its that the pressure torque is zero.

Thanks. We agree and corrected.

line 214 : 'In the presence'

Corrected

References

Abernathey, R., J. Marshall, and D. Ferreira, 2011: The dependence of Southern Ocean meridional overturning on wind stress. J. Phys. Oceanogr., 41, 2261–2278.

Jackson, L., C. W. Hughes, and R. G. Williams, 2006: Topographic control of basin and channel flows: The role of bottom pressure torques and friction. J. Phys. Oceanogr., 36, 1786–1805.

Patmore, R. D., P. R. Holland, D. R. Munday, A. C. Naveira Garabato, D. P. Stevens, and M. P. Meredith, 2019: Topographic control of Southern Ocean gyres and the Antarctic Circumpolar Current: A barotropic perspective. J. Phys. Oceanogr., 49, 3221–3244, doi:10.1175/JPO–D–19–0083.1.

Tansley, C. E. and D. P. Marshall, 2001: On the dynamics of wind-driven circumpolar currents. J. Phys. Oceanogr., 31, 3258–3273.

---

## Author Response (AR2)

**Response to anonymous referee #2 – 18th of August**

In this revised manuscript the authors have done a good job at responding to the comments of both myself and Reviewer 1. The paper remains enjoyable and the revisions have improved its clarity. Where they have declined to act upon a reviewers's comment, the authors have given a good explanation in their response. I have added a few minor comments below that the authors may wish to address. However, there is no reason why these would require another round of review.

There are some interesting questions remaining, which I hope to see the authors take up in future publications. In particular, it would be interesting to explore the how roughness characteristics, such as the variance of the bottom height and horizontal scale, interact with the dynamics of the ACC. The authors have concentrated on eddy saturation in this paper, but I see potential for bottom roughness to also impact upon eddy compensation (reduced sensitivity of the residual MOC to wind stress). Whether sufficient roughness can offset the need for high bathymetry in eddy saturation scenarios would also be interesting, given some of the results of their earlier work (Jouanno et al., 2016).

We acknowledge the reviewer for the time taken to perform this second round of review. We follow the suggestions. We also acknowledge the reviewer for the encouragements to pursue our efforts on this topic.

Minor Comments

lines 82, 83, 107, etc : As noted by Reviewer 1, in a few places the authors continue to use notation of the form 2.10-4, rather than the standard form of 2x10-4.

We corrected the notation here and elsewhere (6 occurrences)

line 125 : "with p_b the pressure", don't need the with at this point.

Corrected.

line 140-144 : The transport of Abernathey & Cessi's (2014) flat bottomed configuration is mentioned here, but not that of their configuration with a ridge (~70Sv, from eyeballing their Figure 8). Other model with ridges produce similar transports, ~90Sv, such as those in Munday et al. (2015) and Marshall et al. (2017).

Thanks, we add these references and corresponding transports.

line 161 : I would say that bottom roughness forces near zero flow at the bottom. In Figure 4 we can see that it isn't quite zero. I imagine there is potential for quite a lot of spatial variation in the bottom flow too. It might be worth reinforcing that it is both the mean and eddying flow that is near zero.

We agree and corrected the sentence.

line 268 : confined TO THE east

Corrected.

line 284 : manifestation OF baroclinic instability

Corrected.

line 286 : mean state leadS to negative

Corrected

line 290-296 : An important caveat to mention regarding Constantinou et al. (2019) is that the regime in which their results indicate barotropic saturation is unlikely to apply to the real Southern Ocean.

In their set of simulations, barotropic saturation occurs for regime II, with wind stress of order $0.1$ N m$^{-2}$ which corresponds to observed wind stress in the Southern Ocean. It therefore seems delicate to us to challenge this study just on the basis of this argument.

Section 4.2 : An extremely relevant reference to this section is Youngs et al. (2017). This paper looks at the energetics of meanders in great detail, highlighting that barotropic energy (conversions) are important to their dynamics and that baroclinic instability is insufficient to do so.

Thank you for pointing out this study. We completed Section 4.2 with the following sentence: "The analysis of the energetics of an ACC standing meander in Youngs et al. (2017) reveals that barotropic instability plays a leading role in the energy

budget of the meander and suggests that baroclinic conversion alone is insufficient to describe both the stratification and distribution of EKE."

lines 326-333 : The experiments of Marshall et al. (2017), which the authors cite elsewhere, also investigate the role of bottom friction and its role in eddy saturation.

Thanks, we now mention that a similar result was observed in Marshall et al. (2017).

References

Jouanno, J., X. Capet, G. Madec, G. Roullet, and P. Klein, 2016: Dissipation of the energy imparted by mid-latitude storms in the southern ocean. Ocean Sci., 12, 743–769, doi:10/5194/os–12–743–2016.

Munday, D. R., H. L. Johnson, and D. P. Marshall, 2015: The role of ocean gateways in the dynamics and sensitivity to wind stress of the early Antarctic Circumpolar Current. Paleoceanography, 30, 284–302, doi:10.1002/2014PA002675.

Youngs, M. K., A. F. Thompson, A. Lazar, and K. J. Richards, 2017: ACC meanders, energy transfer and mixed barotropic-baroclinic instability. J. Phys. Oceanogr., 47, 1291–1305, doi:10.1175/JPO–D–16–0160.1.